# Online Classification with Predictions

**Vinod Raman**
Department of Statistics
University of Michigan
Ann Arbor, MI 48104
vkraman@umich.edu

**Ambuj Tewari**
Department of Statistics
University of Michigan
Ann Arbor, MI 48104
tewaria@umich.edu

## Abstract

We study online classification when the learner has access to predictions about future examples. We design an online learner whose expected regret is never worse than the worst-case regret, gracefully improves with the quality of the predictions, and can be significantly better than the worst-case regret when the predictions of future examples are accurate. As a corollary, we show that if the learner is always guaranteed to observe data where future examples are easily predictable, then online learning can be as easy as transductive online learning. Our results complement recent work in online algorithms with predictions and smoothed online classification, which go beyond a worse-case analysis by using machine-learned predictions and distributional assumptions respectively.

## 1 Introduction

In online classification, Nature plays a game with a learner over $T \in \mathbb{N}$ rounds. In each round $t \in [T]$, Nature selects a labeled example $(x_t, y_t) \in \mathcal{X} \times \mathcal{Y}$ and reveals just the example $x_t$ to the learner. The learner, using the history of the game $(x_1, y_1), ..., (x_{t-1}, y_{t-1})$ and the current example $x_t$, makes a potentially randomized prediction $\hat{y}_t \in \mathcal{Y}$. Finally, Nature reveals the true label $y_t$ and the learner suffers the loss $\mathbb{1}\{\hat{y}_t \neq y_t\}$. Given access to a *hypothesis class* $\mathcal{H} \subseteq \mathcal{Y}^{\mathcal{X}}$ consisting of functions $h : \mathcal{X} \to \mathcal{Y}$, the goal of the learner is to minimize its *regret*, the difference between its cumulative mistake and that of the best fixed hypothesis $h \in \mathcal{H}$ in hindsight. We say a class $\mathcal{H}$ is online learnable if there exists a learning algorithm that achieves vanishing average regret for *any*, potentially adversarial chosen, stream of labeled examples $(x_1, y_1), ..., (x_T, y_T)$. Canonically, one also distinguishes between the realizable and agnostic settings. In the realizable setting, Nature must choose a stream $(x_1, y_1), ..., (x_T, y_T)$ such that there exists a $h \in \mathcal{H}$ for which $h(x_t) = y_t$ for all $t \in [T]$. On the other hand, in the agnostic setting, no such assumptions on the stream are placed.

Due to applications in spam filtering, image recognition, and language modeling, online classification has had a long, rich history in statistical learning theory. In a seminal work, Littlestone [1987] provided a sharp quantitative characterization of which binary hypothesis classes $\mathcal{H} \subseteq \{0, 1\}^{\mathcal{X}}$ are online learnable in the realizable setting. This characterization was in terms of the finiteness of a combinatorial dimension called the Littlestone dimension. Twenty-two years later, Ben-David et al. [2009] proved that the Littlestone dimension continues to characterize the online learnability of binary hypothesis classes in the agnostic setting. Later, Daniely et al. [2011] generalized the Littlestone dimension to multiclass hypothesis classes $\mathcal{H} \subseteq \mathcal{Y}^{\mathcal{X}}$, and showed that it fully characterizes multiclass online learnability when the label space $\mathcal{Y}$ is finite. More recently, Hanneke et al. [2023] extended this result to show that the multiclass Littlestone dimension continues to characterize multiclass online learnability even when $\mathcal{Y}$ is unbounded.

While elegant, the characterization of online classification in terms of the Littlestone dimension is often interpreted as an *impossibility* result [Haghtalab, 2018]. Indeed, due to the restrictive nature of the Littlestone dimension, even simple classes like the 1-dimensional thresholds $\mathcal{H}_{\text{thresh}} = \{x \mapsto$

38th Conference on Neural Information Processing Systems (NeurIPS 2024).

$\mathbb{1}\{x \geq a\} : a \in \mathbb{N}\}$ are not online learnable in the realizable setting. This hardness arises mainly due to a worst-case analysis: the adversary is allowed to choose *any* sequence of labeled examples, even possibly adapting to the learner's strategy. In many situations, however, the sequence of data is "easy" and a worst-case analysis is too pessimistic. For example, if one were to use the daily temperatures to predict snowfall, it is unlikely that temperatures will vary rapidly within a given week. Even so, one might have to access to temperature forecasting models that can accurately predict future temperatures. This motivates a *beyond-worst-case* analysis of online classification algorithms by proving guarantees that adapt to the "easiness" of the example stream.

The push for a beyond-worst-case analysis has its roots in classical algorithm design [Roughgarden, 2021]. Of recent interest is Algorithms with Predictions (AwP), a specific sub-field of beyond-worst-case analysis of algorithms [Mitzenmacher and Vassilvitskii, 2022]. Here, classical algorithms are given additional information about the problem instance in the form of machine-learned predictions. Augmented with these predictions, the algorithm's goal is to perform optimally on a per-input basis when the predictions are good (known as *consistency*), while always ensuring optimal worst-case guarantees (known as *robustness*). Ideally, algorithms are also *smooth*, obtaining performance guarantees that interpolate between instance and worst-case optimality as a function of prediction quality. After a successful application to learning index structures [Kraska et al., 2018], there has been an explosion of work designing algorithms whose guarantees depend on the quality of available, machine-learned predictions Mitzenmacher and Vassilvitskii [2022]. For example, machine-learned predictions have been used to achieve more efficient data-structures [Lin et al., 2022], faster runtimes [Chen et al., 2022, Ergun et al., 2021], better accuracy-space tradeoffs for streaming algorithms [Hsu et al., 2019], and improved performance bounds for online algorithms [Purohit et al., 2018].

Despite this vast literature, the accuracy benefits of machine-learned predictions for online classification are, to the best of our knowledge, unknown. In this work, we bridge the gap between AwP and online classification. In contrast to previous work, which go beyond a worst-case analysis in online classification through smoothness or other distributional assumptions [Haghtalab et al., 2020, Block et al., 2022, Wu et al., 2023], we give the learner access to a *Predictor*, a forecasting algorithm that predicts future *examples* in the data stream. The learner, before predicting a label $\hat{y}_t$, can query the Predictor and receive predictions $\hat{x}_{t+1}, ..., \hat{x}_T$ on the future examples. The learner can then use the history of the game $(x_1, y_1), ..., (x_{t-1}, y_{t-1})$, the current example $x_t$, and the predictions $\hat{x}_{t+1}, ..., \hat{x}_T$ to output a label $\hat{y}_t$. We allow Predictors to be *adaptive* - they can change their predictions of future examples based on the actual realizations of past examples. From this perspective, Predictors are also online learners, and we quantify the *predictability* of example streams through their mistake-bounds.

In this work, we seek to design online learning algorithms whose expected regret, given black-box access to a Predictor, degrades gracefully with the quality of the Predictor's predictions. By doing so, we are also interested in understanding how access to a Predictor may impact the *characterization* of online learnability. In particular, given a Predictor, when can online learnability become *easier* than in the standard, worst-case setup? Guided by these objectives, we make the following contributions.

(1) In the realizable and agnostic settings, we design online learners that, using black-box access to a Predictor, adapt to the "easiness" of the example stream. When the predictions of the Predictor are good, our learner's expected mistakes/regret significantly improves upon the worst-case guarantee. When the Predictor's predictions are bad, the expected mistakes/regret of our learner matches the optimal worst-case expected mistake-bound/regret. Finally, our learner's expected mistake-bound/regret degrades gracefully with the quality of the Predictor's predictions.

(2) We show that having black-box access to a good Predictor can make learning much easier than the standard, worst-case setting. More precisely, good Predictors allow "offline" learnable classes to become online learnable. In this paper, we take the "offline" setting to be transductive online learning [Ben-David et al., 1997, Hanneke et al., 2024] where Nature reveals the entire sequence of examples $x_1, ..., x_T$ (but not the labels $y_1, ..., y_T$) to the learner *before* the game begins. Many "offline" learnable classes are not online learnable. For example, when $\mathcal{Y} = \{0, 1\}$, transductive online learnability is characterized by the finiteness of the VC dimension, the same dimension that characterizes PAC learnability. Thus, our result is analogous to that in smoothed online classification, where PAC learnability is also sufficient for online learnability [Haghtalab et al., 2020, Block et al., 2022].

A notable property of our realizable and agnostic online learners is their use of black-box access to a transductive online learner to make predictions. In this sense, our proof strategies involve reducing online classification with predictions to transductive online learning. For both contributions (1) and (2), we consider only the realizable setting in the main text. The results and arguments for the agnostic setting are nearly identical and thus deferred to Appendix F.

## 1.1 Related Works

**Online Algorithms with Predictions.** Online Algorithms with Predictions (OAwP) has emerged as an important paradigm lying at the intersection of classical online algorithm design and machine learning. Many fundamental online decision-making problems including ski rental [Gollapudi and Panigrahi, 2019, Wang et al., 2020, Bamas et al., 2020], online scheduling [Lattanzi et al., 2020, Wei and Zhang, 2020, Scully et al., 2021], online facility location [Almanza et al., 2021, Jiang et al., 2021], caching [Lykouris and Vassilvitskii, 2021, Elias et al., 2024], and metrical task systems [Antoniadis et al., 2023], have been analyzed under this framework. Recently, Elias et al. [2024] consider a model where the predictor is allowed to learn and adapt its predictions based on the observed data. This is contrast to previous work on learning-augmented online algorithms, where predictions are made from machine learning models trained on historical data, and thus their predictions are static and non-adaptive to the current task at hand. Elias et al. [2024] study a number of fundamental problems, like caching and scheduling, and show how explicitly designed predictors can lead to improved performance bounds. In this work, we consider a model similar to Elias et al. [2024], where the predictions available to the learning algorithms are not fixed, but rather adapt to the true sequence of data processed by the learning algorithm. However, unlike Elias et al. [2024], we do not hand-craft these predictions, but rather assume our learning algorithms have black-box access to a machine-learned prediction algorithm.

**Transductive Online Learning.** In the Transductive Online Learning setting, Nature reveals the entire sequence of examples $x_1, ..., x_T$ to the learner *before* the game begins. The goal of the learner is to predict the corresponding labels $y_1, ..., y_T$ in order, receiving the true label $y_t$ only after making the prediction $\hat{y}_t$ for example $x_t$. First studied by Ben-David et al. [1997], recent work by Hanneke et al. [2024] has established the minimax rates on expected mistakes/regret in the realizable/agnostic settings. In the context of online classification with predictions, one can think of the transductive online learning setting as a special case where the Predictor never makes mistakes.

**Smoothed Online Classification.** In addition to AwP, smoothed analysis [Spielman and Teng, 2009] is another important sub-field of beyond-worst-case analysis of algorithms. By placing distributional assumptions on the input, one can typically go beyond computational and information-theoretic bottlenecks due to worst-case inputs. To this end, Rakhlin et al. [2011], Haghtalab [2018], Haghtalab et al. [2020], Block et al. [2022] consider a *smoothed* online classification model. Here, the adversary has to choose and draw examples from sufficiently anti-concentrated distributions. For binary classification, Haghtalab [2018] and Haghtalab et al. [2020] showed that smoothed online learnability is as easy as PAC learnability. That is, the finiteness of a *smaller* combinatorial parameter called the VC dimension is sufficient for smoothed online classification. In this work, we also go beyond the worst-case analysis standard in online classification, but consider a different model where the adversary is constrained to reveal a sequence of examples that are *predictable*. In this model, we also show that the VC dimension can be sufficient for online learnability.

## 2 Preliminaries

Let $\mathcal{X}$ denote an example space and $\mathcal{Y}$ denote the label space. We make no assumptions about $\mathcal{Y}$, so it can be unbounded (e.g., $\mathcal{Y} = \mathbb{N}$). Let $\mathcal{H} \subseteq \mathcal{Y}^{\mathcal{X}}$ denote a hypothesis class. For a set $A$, let $A^{\star} = \bigcup_{n=0}^{\infty} A^n$ denote the set of all finite sequences of elements in $A$. Moreover, we let $A^{\leq n}$ denote the set of all sequences of elements in $A$ of size at most $n$. Then, $\mathcal{X}^{\star}$ denotes the set of all finite sequences of examples in $\mathcal{X}$ and $\mathcal{Z} \subseteq \mathcal{X}^{\star}$ denotes a particular family of such sequences. We abbreviate a sequence $z_1, ..., z_T$ by $z_{1:T}$. Finally, for $a, b, c \in \mathbb{R}$, we let $a \wedge b \wedge c = \min\{a, b, c\}$.

## 2.1 Online Classification

In online classification, a learner $\mathcal{A}$ plays a repeated game against Nature over $T \in \mathbb{N}$ rounds. In each round $t \in [T]$, Nature picks a labeled example $(x_t, y_t) \in \mathcal{X} \times \mathcal{Y}$ and reveals $x_t$ to the learner. The learner makes a randomized prediction $\hat{y}_t \in \mathcal{Y}$. Finally, Nature reveals the true label $y_t$ and the learner suffers the 0-1 loss $\mathbb{1}\{\hat{y}_t \neq y_t\}$. Given a hypothesis class $\mathcal{H} \subseteq \mathcal{Y}^{\mathcal{X}}$, the goal of the learner is to minimize its *expected regret*

$$\mathrm{R}_{\mathcal{A}}(T, \mathcal{H}) := \sup_{x_{1:T} \in \mathcal{X}} \sup_{y_{1:T} \in \mathcal{Y}^T} \left( \mathbb{E}\left[ \sum_{t=1}^{T} \mathbb{1}\{\mathcal{A}(x_t) \neq y_t\} \right] - \inf_{h \in \mathcal{H}} \sum_{t=1}^{T} \mathbb{1}\{h(x_t) \neq y_t\} \right),$$

where the expectation is only over the randomness of the learner. A hypothesis class $\mathcal{H}$ is said to be online learnable if there exists an (potentially randomized) online learning algorithm $\mathcal{A}$ such that $\mathrm{R}_{\mathcal{A}}(T, \mathcal{H}) = o(T)$. If it is guaranteed that the learner always observes a sequence of examples labeled by some hypothesis $h \in \mathcal{H}$, then we say we are in the *realizable* setting and the goal of the learner is to minimize its *expected cumulative mistakes*,

$$\mathrm{M}_{\mathcal{A}}(T, \mathcal{H}) := \sup_{x_{1:T} \in \mathcal{X}^T} \sup_{h \in \mathcal{H}} \mathbb{E}\left[ \sum_{t=1}^{T} \mathbb{1}\{\mathcal{A}(x_t) \neq h(x_t)\} \right],$$

where again the expectation is taken only with respect to the randomness of the learner. It is well known that the finiteness of the multiclass extension of the Littlestone dimension (Ldim) characterizes realizable and agnostic online learnability [Littlestone, 1987, Daniely et al., 2011, Hanneke et al., 2023]. See Appendix A for complete definitions.

## 2.2 Online Classification with Predictions

Motivated by the fact that real-world example streams $x_{1:T}$ are far from worst-case, we give our learner $\mathcal{A}$ black-box access to a *Predictor* $\mathcal{P}$, defined algorithmically in Algorithm 1 and formally in Definition 1. In the rest of the paper, we abuse notation by not explicitly indicating that $\mathcal{P}$ takes its own past predictions as input. That is, given a sequence $x_{1:T} \in \mathcal{X}^T$, we will let $\mathcal{P}(x_{1:t})$ denotes its prediction on the $t$'th round.

**Definition 1** (Predictor). *A Predictor $\mathcal{P} : (\mathcal{X} \times \mathcal{X}^T)^\star \to \Pi(\mathcal{X}^T)$ is a map that takes in a sequence of instances $x_1, x_2, ...$, its own past predictions $\hat{x}^1_{1:T}, \hat{x}^2_{1:T}, ...$, and outputs a distribution $\hat{\mu} \in \Pi(\mathcal{X}^T)$. The Predictor make its next prediction by sampling $\hat{x}_{1:T} \sim \hat{\mu}$.*

---

**Algorithm 1** Predictor $\mathcal{P}$

---

**Input:** Time horizon $T$

1 **for** $t = 1, ..., T$ **do**
2       Nature reveals the true example $x_t$.
3       Observe $x_t$, update, and make a (potentially randomized) prediction $\hat{x}^t_{1:T}$.
4 **end**

---

**Remark.** We highlight that our Predictors are very general and can also use side information, in addition to the past examples, to make predictions about future examples. For example, if the examples are daily average temperatures, then Predictors can also use other covariates, like humidity, precipitation, and wind speed, to predict future temperatures.

In each round $t \in [T]$, the learner $\mathcal{A}$ can query the Predictor $\mathcal{P}$ to get a sense of what examples it will observe in the future. Then, the learner $\mathcal{A}$ can use the history $(x_1, y_1), .., (x_{t-1}, y_{t-1})$, the current example $x_t$, *and* the future predicted examples to classify the current example. Protocol 2 makes explicit the interaction between the learner, the Predictor, and Nature.

Note that, in every round $t \in [T]$, the Predictor $\mathcal{P}$ makes a prediction about the *entire* sequence of $T$ examples, even those that it has observed in the past. This is mainly for notational convenience as we assume that our Predictors are *consistent*.

**Assumption 1** (Consistency). *A Predictor is* consistent *if for every sequence $x_{1:T} \in \mathcal{X}^T$ and every time point $t \in [T]$, the prediction $\hat{x}_{1:T} = \mathcal{P}(x_{1:t})$ satisfies the property that $\hat{x}_{1:t} = x_{1:t}$.*

---

**Algorithm 2** Online Learning with a Predictor

---

**Input:** Predictor $\mathcal{P}$, Hypothesis class $\mathcal{H}$, Time horizon $T$

1 **for** $t = 1, ..., T$ **do**
2     Nature reveals the true example $x_t$.
3     The Predictor $\mathcal{P}$ observes $x_t$, updates, and reveals its predictions $\hat{x}^t_{1:T}$.
4     Learner makes a randomized prediction $\hat{y}_t$ using $\hat{x}^t_{1:T}, x_t$, and $(x_1, y_1), ..., (x_{t-1}, y_{t-1})$.
5     Nature reveals the true label $y_t$ to the learner.
6     Learner suffers loss $\mathbb{1}\{y_t \neq \hat{y}_t\}$ and updates itself.
7 **end**

---

Although stated as an assumption, it is without loss of generality that Predictors are consistent - any inconsistent Predictor can be made consistent by hard coding its input into its output. In addition to consistency, we assume that our Predictors are *lazy*.

**Assumption 2** (Laziness). *A Predictor is* lazy *if for every sequence $x_{1:T} \in \mathcal{X}^T$ and every $t \in [T]$, if $\mathcal{P}(x_{1:t-1})_t = x_t$, then $\mathcal{P}(x_{1:t}) = \mathcal{P}(x_{1:t-1})$. That is, $\mathcal{P}$ does not change its prediction if it is correct.*

Since Predictors are also online learners, the assumption of laziness is also mild: non-lazy online learners can be generically converted into lazy ones [Littlestone, 1987, 1989]. We always assume that Predictors are consistent and lazy and drop these pronouns for the rest of the paper.

**Remark.** We highlight that Predictors are adaptive and change their predictions based on the realizations of past examples. This is contrast to existing literature in OAwP, where machine-learned predictions are often static. Nevertheless, our framework is more general and captures the setting where predictions of examples are made once and fixed throughout the game. Indeed, consider the consistent, lazy Predictor that fixes a sequence $z_{1:T} \in \mathcal{X}^T$ before the game begins, and for every $t \in [T]$, outputs the predictions $\hat{x}^t_{1:T}$ such that $\hat{x}^t_{1:t} = x_{1:t}$ and $\hat{x}^t_{t+1:T} = z_{t+1:T}$.

Ideally, when given access to a Predictor $\mathcal{P}$, the expected regret of $\mathcal{A}$ should degrade gracefully with the quality of $\mathcal{P}$'s predictions. To this end, we quantify the performance of a Predictor $\mathcal{P}$ through

$$\mathrm{M}_{\mathcal{P}}(x_{1:T}) := \mathbb{E}\left[\sum_{t=2}^{T} \mathbb{1}\{\mathcal{P}(x_{1:t-1})_t \neq x_t\}\right],$$

the expected number of mistakes that $\mathcal{P}$ makes on a sequence of examples $x_{1:T} \in \mathcal{X}^T$. In Section 3, we design an online learner whose expected regret/mistake-bound on a stream $(x_1, y_1), ..., (x_T, y_T)$ can be written in terms of $\mathrm{M}_{\mathcal{P}}(x_{1:T})$.

### 2.3 Predictability

Predictors and their mistake bounds offer us to ability to define and quantify a notion of "easiness" for example streams $x_{1:T}$. In particular, we can distinguish between example streams that are predictable and unpredictable. To do so, let $\mathcal{Z} \subseteq \mathcal{X}^\star$ denote a collection of finite sequences of examples. By restricting Nature to playing examples streams in $\mathcal{Z}$, we can define analogous notions of minimax expected regret

$$\mathrm{R}_{\mathcal{A}}(T, \mathcal{H}, \mathcal{Z}) := \sup_{x_{1:T} \in \mathcal{Z}} \sup_{y_{1:T} \in \mathcal{Y}^T} \mathbb{E}\left[\sum_{t=1}^{T} \mathbb{1}\{\mathcal{A}(x_t) \neq y_t\} - \inf_{h \in \mathcal{H}} \sum_{t=1}^{T} \mathbb{1}\{h(x_t) \neq y_t\}\right],$$

and minimax expected mistakes,

$$\mathrm{M}_{\mathcal{A}}(T, \mathcal{H}, \mathcal{Z}) := \sup_{x_{1:T} \in \mathcal{Z}} \sup_{h \in \mathcal{H}} \mathbb{E}\left[\sum_{t=1}^{T} \mathbb{1}\{\mathcal{A}(x_t) \neq h(x_t)\}\right].$$

As usual, we say that a tuple $(\mathcal{H}, \mathcal{Z})$ is online and realizable online learnable if $\inf_{\mathcal{A}} \mathrm{R}_{\mathcal{A}}(T, \mathcal{H}, \mathcal{Z}) = o(T)$ and $\inf_{\mathcal{A}} \mathrm{M}_{\mathcal{A}}(T, \mathcal{H}, \mathcal{Z}) = o(T)$ respectively. If $\mathcal{Z} = \mathcal{X}^\star$, then the definitions above recover the standard, worst-case online classification setup. However, in the more general case where $\mathcal{Z}^\star \subseteq \mathcal{X}^\star$, we can use the *existence* of good Predictors $\mathcal{P}$ and their mistake bounds to quantify the "easiness" of a stream class $\mathcal{Z}$. That is, we say $\mathcal{Z}$ is predictable if there exists a consistent, lazy Predictor $\mathcal{P}$ such that $\mathrm{M}_{\mathcal{P}}(T, \mathcal{Z}) := \sup_{x_{1:T} \in \mathcal{Z}} \mathrm{M}_{\mathcal{P}}(x_{1:T}) = o(T)$.

**Definition 2** (Predictability). *A class $\mathcal{Z} \subseteq \mathcal{X}^\star$ is predictable if and only if $\inf_{\mathcal{P}} \mathrm{M}_{\mathcal{P}}(T, \mathcal{Z}) = o(T)$.*

Definition 2 provides a qualitative definition of what it means for a sequence of examples to be predictable, and therefore "easy". If $\mathcal{Z} \subseteq \mathcal{X}^\star$ is a predictable class of example streams, then a stream $x_{1:T} \in \mathcal{X}^T$ is predictable if $x_{1:T} \in \mathcal{Z}$. By having access to a good Predictor, sequences of examples that previously exhibited "worst-case" behavior, now become predictable. One natural predictable collection of streams are those induced by easy-to-learn discrete-time dynamical systems [Raman et al., 2024]. That is, let $\mathcal{X}$ be the state space for a finite collection $\mathcal{G}$ of transition functions. Then, given an initial state $x_0 \in \mathcal{X}$, one can consider the stream class $\mathcal{Z}$ to be the set of all trajectories induced by transition functions in $\mathcal{G}$. In Section 3, we show that for such classes of predictable examples, "offline" learnability is sufficient for online learnability.

## 2.4 Offline Learnability

In the classical analysis of online algorithms, one competes against the best "offline" solution. In the context of online classification, this amounts to comparing online learnability to "offline" learnability, where we interpret the "offline" setting as the case where Nature reveals the sequence of examples $(x_1, ..., x_T)$ before the game begins. In particular, compared to the standard online learning setting, in the "offline" version, the learner knows the sequence of examples $x_1, ..., x_T$ before the game begins, and its goal is to predict the corresponding labels $y_1, ..., y_T$. This setting was recently named "Transductive Online Learning" [Hanneke et al., 2024] and the minimax rates in both the realizable and agnostic setting have been established [Ben-David et al., 1997, Hanneke et al., 2023]. For the remainder of the paper, we will use offline and transductive online learnability interchangeably.

For a randomized offline learner $\mathcal{B}$, we let

$$\mathrm{R}_{\mathcal{B}}(T, \mathcal{H}) := \sup_{x_{1:T} \in \mathcal{X}^T} \sup_{y_{1:T} \in \mathcal{Y}^T} \mathbb{E}\left[\sum_{t=1}^T \mathbb{1}\{\mathcal{B}_{x_{1:T}}(x_t) \neq y_t\} - \inf_{h \in \mathcal{H}} \sum_{t=1}^T \mathbb{1}\{h(x_t) \neq y_t\}\right]$$

denote its minimax expected regret and

$$\mathrm{M}_{\mathcal{B}}(T, \mathcal{H}) := \sup_{h \in \mathcal{H}} \sup_{x_{1:T} \in \mathcal{X}^T} \mathbb{E}\left[\sum_{t=1}^T \mathbb{1}\{\mathcal{B}_{x_{1:T}}(x_t) \neq h(x_t)\}\right].$$

denote its minimax expected mistakes. We use the notation $\mathcal{B}_{x_{1:T}}$ to indicate that $\mathcal{B}$ was initialized with the sequence $x_{1:T}$ before the game begins. If $\mathrm{M}_{\mathcal{B}}(T, \mathcal{H}) = o(T)$ or $\mathrm{R}_{\mathcal{B}}(T, \mathcal{H}) = o(T)$, then we say that $\mathcal{B}$ is a no-regret offline learner. It turns out that realizable and agnostic offline learnability are equivalent [Hanneke et al., 2024]. That is, $\mathrm{M}_{\mathcal{B}}(T, \mathcal{H}) = o(T) \Leftrightarrow \mathrm{R}_{\mathcal{B}}(T, \mathcal{H}) = o(T)$. Thus, we say a class $\mathcal{H} \subseteq \mathcal{Y}^{\mathcal{X}}$ is offline learnable if and only if there exists a no-regret offline learner for $\mathcal{H}$.

When $|\mathcal{Y}| = 2$, Ben-David et al. [1997] and Hanneke et al. [2023] show that the finiteness of a combinatorial dimension called the Vapnik–Chervonenkis (VC) dimension (or equivalently PAC learnability) is sufficient for offline learnability (see Appendix A for complete definitions).

**Lemma 2.1** (Ben-David et al. [1997], Hanneke et al. [2024]). *For every $\mathcal{H} \subseteq \{0,1\}^{\mathcal{X}}$, there exists a deterministic offline learner $\mathcal{B}$ such that*

$$\mathrm{M}_{\mathcal{B}}(T, \mathcal{H}) = O\left(\mathrm{VC}(\mathcal{H}) \log_2 T\right)$$

*where $\mathrm{VC}(\mathcal{H})$ is the VC dimension of $\mathcal{H}$.*

In Section 3, we use this upper bound in Lemma 2.1 to prove that PAC learnability of $\mathcal{H}$ implies $(\mathcal{H}, \mathcal{Z})$ online learnability when $\mathcal{Z}$ is predictable. Interestingly, Hanneke et al. [2024] also establish a trichotomy in the minimax expected mistakes for offline learning in the realizable setting. That is, for any $\mathcal{H} \subseteq \mathcal{Y}^{\mathcal{X}}$ with $|\mathcal{Y}| < \infty$, the quantity $\mathrm{M}_{\mathcal{B}}(T, \mathcal{H})$ can only be $\Theta(1)$, $\Theta(\log_2 T)$, or $\Theta(T)$. On the other hand, in the agnostic setting, $\mathrm{R}_{\mathcal{B}}(T, \mathcal{H})$ can be $\tilde{\Theta}(\sqrt{T})$ or $\Theta(T)$, where $\tilde{\Theta}$ hides poly-log terms in $T$.

Our main result in Section 3 shows that offline learnability is sufficient for online learnability under predictable examples. The following technical lemma will be important when proving so.

**Lemma 2.2.** [Ceccherini-Silberstein et al., 2017, Lemma 5.17] *Let $g : \mathbb{Z}_+ \mapsto \mathbb{R}_+$ be a positive sublinear function. Then, $g$ is bounded from above by a concave sublinear function $f : \mathbb{R}_+ \mapsto \mathbb{R}_+$.*

In light of Lemma 2.2, we let $\bar{f}$ denote the smallest concave sublinear function upper bounding the positive sublinear function $f$. For example, our regret bounds in Section 3 will often be in terms of $\overline{\mathrm{M}}_{\mathcal{B}}(T, \mathcal{H})$. Although in full generality $\mathrm{M}_{\mathcal{B}}(T, \mathcal{H}) \leq \overline{\mathrm{M}}_{\mathcal{B}}(T, \mathcal{H})$, in many cases we have equality. For example, when $|\mathcal{Y}| = 2$, the trichotomy of expected minimax rates established by Theorem 4.1 in Hanneke et al. [2024] shows that $\mathrm{M}_{\mathcal{B}}(T, \mathcal{H}) = \overline{\mathrm{M}}_{\mathcal{B}}(T, \mathcal{H})$.

# 3 Adaptive Rates in the Realizable Setting

In this section, we design learning algorithms whose expected mistake bounds, given black-box access to a Predictor $\mathcal{P}$ and offline learner $\mathcal{B}$, adapt to the quality of predictions by $\mathcal{P}$ and $\mathcal{B}$. Our main quantitative result is stated below.

**Theorem 3.1** (Realizable upper bound). *For every $\mathcal{H} \subseteq \mathcal{Y}^{\mathcal{X}}$, Predictor $\mathcal{P}$, and no-regret offline learner $\mathcal{B}$, there exists an online learner $\mathcal{A}$ such that for every realizable stream $(x_1, y_1), ..., (x_T, y_T)$, $\mathcal{A}$ makes at most*

$$3\left( \underbrace{\mathrm{L}(\mathcal{H})}_{(i)} \wedge \underbrace{(\mathrm{M}_{\mathcal{P}}(x_{1:T}) + 1)\, \mathrm{M}_{\mathcal{B}}(T, \mathcal{H})}_{(ii)} \wedge \underbrace{6\Big((\mathrm{M}_{\mathcal{P}}(x_{1:T}) + 1)\, \overline{\mathrm{M}}_{\mathcal{B}}\Big(\frac{T}{\mathrm{M}_{\mathcal{P}}(x_{1:T}) + 1} + 1, \mathcal{H}\Big) + \log_2 T\Big)}_{(iii)} \right) + 5$$

*mistakes in expectation, where $\mathrm{L}(\mathcal{H})$ is the Littlestone dimension of $\mathcal{H}$.*

We highlight some important consequences of Theorem 3.1. Firstly, when $\mathrm{M}_{\mathcal{P}}(x_{1:T}) = 0$, the expected mistake bound of $\mathcal{A}$ matches (up to constant factors) that of the offline learner $\mathcal{B}$. Thus, when $\mathrm{M}_{\mathcal{P}}(x_{1:T}) = 0$ and $\mathcal{B}$ is a minimax optimal offline learner, our learner $\mathcal{A}$ performs as well as the best offline learner. Secondly, the expected mistake bound of $\mathcal{A}$ is always at most $3\,\mathrm{L}(\mathcal{H}) + 5$; the minimax worst-case mistake bound up to constant factors. Thus, our learner $\mathcal{A}$ never does worse than the worst-case mistake bound. Thirdly, the expected mistake bound of $\mathcal{A}$ gracefully interpolates between the offline and worst-case optimal rates as a function of $\mathrm{M}_{\mathcal{P}}(x_{1:T})$. In Section 3.3, we show that the dependence of $\mathcal{A}$'s mistake bound on $\mathrm{M}_{\mathcal{P}}(x_{1:T})$ and $\mathrm{M}_{\mathcal{B}}(T, \mathcal{H})$ can be tight. Lastly, we highlight that Theorem 3.1 makes no assumption about the size of $\mathcal{Y}$.

With respect to learnability, Corollary 3.2 shows that offline learnability of $\mathcal{H}$ is sufficient for online learnability under predictable examples.

**Corollary 3.2** (Offline learnability $\implies$ Realizable Online learnability with Predictable Examples). *For every $\mathcal{H} \subseteq \mathcal{Y}^{\mathcal{X}}$ and $\mathcal{Z} \subseteq \mathcal{X}^{\star}$,*

$$\mathcal{Z} \text{ is predictable and } \mathcal{H} \text{ is offline learnable } \implies (\mathcal{H}, \mathcal{Z}) \text{ is realizable online learnable.}$$

This follows from a slight modification of the proof of Theorem 3.1 along with the fact that the term $(\mathrm{M}_{\mathcal{P}}(T, \mathcal{Z}) + 1)\overline{\mathrm{M}}_{\mathcal{B}}\Big(\frac{T}{\mathrm{M}_{\mathcal{P}}(T, \mathcal{Z}) + 1}, \mathcal{H}\Big) = o(T)$ when $\mathrm{M}_{\mathcal{B}}(T, \mathcal{H}) = o(T)$ and $\mathrm{M}_{\mathcal{P}}(T, \mathcal{Z}) = o(T)$. In addition, we can also establish a quantitative version of Corollary 3.2 for VC classes.

**Corollary 3.3.** *For every $\mathcal{H} \subseteq \{0, 1\}^{\mathcal{X}}$, Predictor $\mathcal{P}$ and $\mathcal{Z} \subseteq \mathcal{X}^{\star}$, there exists an online learner $\mathcal{A}$ such that*

$$\mathrm{M}_{\mathcal{A}}(T, \mathcal{H}, \mathcal{Z}) = O\left( \mathrm{VC}(\mathcal{H})(\mathrm{M}_{\mathcal{P}}(T, \mathcal{Z}) + 1) \log_2\Big(\frac{T}{\mathrm{M}_{\mathcal{P}}(T, \mathcal{Z}) + 1}\Big) + \log_2 T \right).$$

We prove both Corollary 3.2 and 3.3 in Appendix C. Corollary 3.3 shows that PAC learnability implies online learnability under predictable examples. Moreover, for VC classes, when $\mathrm{M}_{\mathcal{P}}(x_{1:T}) = 0$, the upper bound in Corollary 3.3 exactly matches that of Lemma 2.1. An analogous corollary in terms of the Natarajan dimension (see Appendix A for definition) holds when $|\mathcal{Y}| < \infty$.

The remainder of this section is dedicated to proving Theorem 3.1. The proof involves constructing three *different* online learners, with expected mistake bounds (i), (ii), and (iii) respectively, and then running the Deterministic Weighted Majority Algorithm (DWMA) using these learners as experts [Arora et al., 2012]. The following guarantee of DWMA along with upper bounds (i), (ii), and (iii) gives the upper bound in Theorem 3.1 (see Appendix D for complete proof).

**Lemma 3.4** (DWMA guarantee [Arora et al., 2012]). *The DWMA run with $N$ experts and learning rate $\eta = 1/2$ makes at most $3(\min_{i \in [N]} M_i + \log_2 N)$ mistakes, where $M_i$ is the number of mistakes made by expert $i \in [N]$.*

The online learner obtaining the upper bound $\mathrm{L}(\mathcal{H})$ is the celebrated Standard Optimal Algorithm [Littlestone, 1987, Daniely et al., 2011], and thus we omit the details here. Our second and third learners are described in Sections 3.1 and 3.2 respectively. Finally, in Section 3.3, we give a lower bound showing that our upper bound in Theorem 3.1 can be tight.

## 3.1 Proof of upper bound (ii) in Theorem 3.1

Consider a lazy, consistent predictor $\mathcal{P}$. Given any sequence of examples $x_{1:T} \in \mathcal{X}^T$, the Predictor $\mathcal{P}$ makes $c \in \mathbb{N}$ mistakes at some timepoints $t_1, ..., t_c \in [T]$. Since $\mathcal{P}$ may be randomized, both $c$ and $t_1, ..., t_c$ are random variables. Crucially, since $\mathcal{P}$ is lazy, for every $i \in \{0, ..., c+1\}$, the predictions made by $\mathcal{P}$ on timepoints strictly between $t_i$ and $t_{i+1}$ are correct and remain unchanged, where we take $t_0 = 0$ and $t_{c+1} = T + 1$. This means that on round $t_i$, we have that $\hat{x}^{t_i}_{t_i:t_{i+1}-1} = x_{t_i:t_{i+1}-1}$. Therefore, initializing a fresh copy of an offline learner $\mathcal{B}$ with the predictions $\hat{x}^{t_i}_{t_i:T}$ ensures that $\mathcal{B}$ makes at most $\mathrm{M}_{\mathcal{B}}(T - t_i + 1, \mathcal{H})$ mistakes on the stream $(x_{t_i}, y_{t_i}), ..., (x_{t_{i+1}-1}, y_{t_{i+1}-1})$. Repeating this argument for all adjacent pairs of timepoints in $\{t_1, ..., t_c\}$, gives the following strategy: initialize a new offline learner $\mathcal{B}$ every time $\mathcal{P}$ makes a mistakes, and use $\mathcal{B}$ to make predictions until the next time $\mathcal{P}$ makes a mistake. Algorithm 3 implements this idea.

---

**Algorithm 3** Online Learner

**Input:** Hypothesis class $\mathcal{H}$, Offline learner $\mathcal{B}$, Time horizon $T$
1 **Initialize:** $i = 0$
2 **for** $t = 1, ..., T$ **do**
3      Receive $x_t$ from Nature.
4      Receive predictions $\hat{x}^t_{1:T}$ from Predictor $\mathcal{P}$ such that $\hat{x}^t_{1:t} = x_{1:t}$.
5      **if** $t = 1$ or $\hat{x}^{t-1}_t \neq x_t$ (i.e. $\mathcal{P}$ made a mistake) **then**
6          Let $\mathcal{B}^i$ be a new copy of $\mathcal{B}$ initialized with the sequence $\hat{x}^t_{t:T}$ and set $i \leftarrow i + 1$.
7      Query $\mathcal{B}^i$ on example $x_t$ and play its returned prediction $\hat{y}_t$.
8      Receive true label $y_t$ from Nature and pass it to $\mathcal{B}^i$.
9 **end**

---

**Lemma 3.5.** *For every $\mathcal{H} \subseteq \mathcal{Y}^{\mathcal{X}}$, Predictor $\mathcal{P}$, no-regret offline learner $\mathcal{B}$, and realizable stream $(x_1, y_1), ..., (x_T, y_T)$, Algorithm 3 makes at most $(\mathrm{M}_{\mathcal{P}}(x_{1:T}) + 1)\,\mathrm{M}_{\mathcal{B}}(T, \mathcal{H})$ mistakes in expectation.*

*Proof.* Let $\mathcal{A}$ denote Algorithm 3, $(x_1, y_1), ..., (x_T, y_T)$ denote the realizable stream to be observed by $\mathcal{A}$, and $h^\star \in \mathcal{H}$ to be the labeling hypothesis. Let $c$ be the random variable denoting the number of mistakes made by Predictor $\mathcal{P}$ on the stream and $t_1, ..., t_c$ be the random variables denoting the time points where $\mathcal{P}$ makes these errors (e.g . $\hat{x}^{t_i-1}_{t_i} \neq x_{t_i}$). Note that $t_i \geq 2$ for all $i \in [c]$. We will show pointwise for every value of $c$ and $t_1, ..., t_c$ that $\mathcal{A}$ makes at most $(c + 1)\,\mathrm{M}_{\mathcal{B}}(T, \mathcal{H})$ mistakes in expectation over the randomness of $\mathcal{B}$. Taking an outer expectation with respect to the randomness of $\mathcal{P}$ and using the fact that $\mathbb{E}\,[c] = \mathrm{M}_{\mathcal{P}}(x_{1:T})$, completes the proof.

First, consider the case where $c = 0$ (i.e. $\mathcal{P}$ makes no mistakes). Then, since $\mathcal{P}$ is lazy, we have that $\hat{x}^t_{1:T} = x_{1:T}$ for every $t \in [T]$. Thus line 5 fires exactly once on round $t = 1$, $\mathcal{A}$ initializes an offline learner $\mathcal{B}^1$ with $x_{1:T}$, and $\mathcal{A}$ uses $\mathcal{B}^1$ to make its prediction on all rounds. Thus, $\mathcal{A}$ makes at most $\mathrm{M}_{\mathcal{B}}(T, \mathcal{H})$ mistakes in expectation.

Now, let $c > 0$ and $t_1, ..., t_c$ be the time points where $\mathcal{P}$ errs. Partition the sequence $1, ..., T$ into the disjoint intervals $(1, ..., t_1 - 1), (t_1, ..., t_2 - 1), ..., (t_c, ..., T)$. Define $t_0 := 1$ and $t_{c+1} := T$. Fix an $i \in \{0, ..., c\}$. Observe that for every $j \in \{t_i, ..., t_{i+1} - 1\}$, we have that $\hat{x}^j_{1:t_{i+1}-1} = x_{t_{i+1}-1}$. This comes from the fact that $\mathcal{P}$ does not error on timepoints $t_i + 1, ..., t_{i+1} - 1$ and is both consistent and lazy (see Assumptions 1 and 2). Thus, line 5 fires on round $t_i$, $\mathcal{A}$ initializes an offline learner $\mathcal{B}^i$ with the sequence $\hat{x}^{t_i}_{t_i:T} = x_{t_i:t_{i+1}-1} \circ \hat{x}^{t_i}_{t_{i+1}:T}$, and $\mathcal{A}$ uses $\mathcal{B}^i$ it to make predictions for all remaining timepoints $t_i, ..., t_{i+1} - 1$. Note that line 5 does not fire on timepoints $t_i + 1, ..., t_{i+1} - 1$.

Consider the hypothetical labeled stream of examples

$$(\hat{x}_{t_i}^{t_i}, h^\star(\hat{x}_{t_i}^{t_i})), ..., (\hat{x}_T^{t_i}, h^\star(\hat{x}_T^{t_i})) = (x_{t_i}, y_{t_i}), ..., (x_{t_{i+1}-1}, y_{t_{i+1}-1}), (\hat{x}_{t_{i+1}}^{t_i}, h^\star(\hat{x}_{t_{i+1}}^{t_i})), ..., (\hat{x}_T^{t_i}, h^\star(\hat{x}_T^{t_i})).$$

By definition, $\mathcal{B}^i$, after initialized with $\hat{x}_{t_i:T}^{t_i}$, makes at most $\mathrm{M}_\mathcal{B}(T - t_i + 1, \mathcal{H})$ mistakes in expectation when simulated on the stream $(\hat{x}_{t_i}^{t_i}, h^\star(\hat{x}_{t_i}^{t_i})), ..., (\hat{x}_T^{t_i}, h^\star(\hat{x}_T^{t_i}))$. Thus, $\mathcal{B}^i$ makes at most $\mathrm{M}_\mathcal{B}(T, \mathcal{H})$ mistakes in expectation on the *prefix* $(\hat{x}_{t_i}^{t_i}, h^\star(\hat{x}_{t_i}^{t_i})), ..., (\hat{x}_{t_{i+1}-1}^{t_i}, h^\star(\hat{x}_{t_{i+1}-1}^{t_i})) = (x_{t_i}, y_{t_i}), ..., (x_{t_{i+1}-1}, y_{t_{i+1}-1})$. Since on the interval timepoint $t_i$, $\mathcal{A}$ instantiates $\mathcal{B}^i$ with the sequence $\hat{x}_{t_i:T}^{t_i}$ and proceeds to simulate $\mathcal{B}^i$ on the sequence of labeled examples $(x_{t_i}, y_{t_i}), ..., (x_{t_{i+1}-1}, y_{t_{i+1}-1})$, $\mathcal{A}$ makes at most $\mathrm{M}_\mathcal{B}(T, \mathcal{H})$ mistakes in expectation on the sequence $(x_{t_i}, y_{t_i}), ..., (x_{t_{i+1}-1}, y_{t_{i+1}-1})$. Since the interval $i$ was chosen arbitrarily, the above analysis is true for every $i \in \{0, ..., c\}$ and therefore $\mathcal{A}$ makes at most $(c + 1) \mathrm{M}_\mathcal{B}(T, \mathcal{H})$ mistakes in expectation over the entire stream. $\qquad\square$

### 3.2 Proof sketch of upper bound (iii) in Theorem 3.1

When $\mathrm{M}_\mathcal{B}(T, \mathcal{H})$ is large (i.e. $\Omega(\sqrt{T})$), upper bound (ii) is sub-optimal. Indeed, if $t_1, ..., t_c$ denotes the timepoints where $\mathcal{P}$ makes mistakes on the stream $x_{1:T}$, then Algorithm 3 initializes offline learners with sequences of length $T - t_i + 1$. The resulting mistake-bound of these offline learners are then in the order of $T - t_i + 1$, which can be large if $t_1, ..., t_c$ are evenly spaced across the time horizon. To overcome this, we construct a *family* $\mathcal{E}$ of online learners, each of which explicitly controls the length of the sequences offline learners can be initialized with. Finally, we run DWMA using $\mathcal{E}$ as its set of experts. Our family of online learners is parameterized by integers $c \in \{0, ..., T - 1\}$. Given an input $c \in \{0, ..., T - 1\}$, the online learner parameterized by $c$ partitions the stream into $c + 1$ roughly equally sized parts of size $\lceil \frac{T}{c+1} \rceil$ and runs a fresh copy of Algorithm 3 on each partition. In this way, the online learner parameterized by $c$ ensures that offline learners are initialized with time horizons at most $\lceil \frac{T}{c+1} \rceil$. Algorithm 4 formalizes this online learner and Lemma 3.6, whose proof is in Appendix B, bounds its expected number of mistakes.

---

**Algorithm 4** Expert$(c)$

---

**Input:** Copy of Algorithm 3 denoted $\mathcal{K}$, Offline Learner $\mathcal{B}$, Time horizon $T$
1 **Initialize:** $\tilde{t}_i = i\lceil \frac{T}{c+1} \rceil$ for $i \in \{1, ..., c\}$, $\tilde{t}_0 = 0$, and $\tilde{t}_{c+1} = T$.
2 **for** $t = 1, ..., T$ **do**
3      Let $i \in \{0, ..., c\}$ such that $t \in \{\tilde{t}_i + 1, ..., \tilde{t}_{i+1}\}$.
4      **if** $t = \tilde{t}_i + 1$ **then**
5         Let $\mathcal{K}_i$ be a new copy of $\mathcal{K}$ initialized with time horizon $\tilde{t}_{i+1} - \tilde{t}_i$ and a new copy of $\mathcal{B}$.
6      Receive $x_t$ from Nature.
7      Receive predictions $\hat{x}_{1:T}^t$ from Predictor $\mathcal{P}$ such that $\hat{x}_{1:t}^t = x_{1:t}$.
8      Forward $x_t$ and $\hat{x}_{\tilde{t}_i+1:\tilde{t}_{i+1}}^t$ to $\mathcal{K}_i$ via Lines 2 and 3 of Algorithm 3 respectively.
9      Receive $\hat{y}_t$ from $\mathcal{K}_i$ via line 6 in Algorithm 3 and predict $\hat{y}_t$.
10      Receive true label $y_t$ and forward it to $\mathcal{K}_i$ via line 7 in Algorithm 3.
11 **end**

---

**Lemma 3.6** (Expert guarantee). *For any $\mathcal{H} \subseteq \mathcal{Y}^\mathcal{X}$, Predictor $\mathcal{P}$, and no-regret offline learner $\mathcal{B}$, Algorithm 4, given as input $c \in \{0, ..., T - 1\}$, makes at most*

$$(\mathrm{M}_\mathcal{P}(x_{1:T}) + c + 1)\overline{\mathrm{M}}_\mathcal{B}\Big(\frac{T}{c + 1} + 1, \mathcal{H}\Big)$$

*mistakes in expectation on any realizable stream $(x_1, y_1), ..., (x_T, y_T)$.*

Note that when $c = 0$ and $\mathrm{M}_\mathcal{B}(T, \mathcal{H}) = \overline{\mathrm{M}}_\mathcal{B}(T, \mathcal{H})$, this bound reduces to the one in Lemma 3.5 up to a constant factor. On the other hand, using $c = \lceil \mathrm{M}_\mathcal{P}(x_{1:T}) \rceil$ gives the upper bound

$$2(\mathrm{M}_\mathcal{P}(x_{1:T}) + 1)\overline{\mathrm{M}}_\mathcal{B}\Big(\frac{T}{\mathrm{M}_\mathcal{P}(x_{1:T}) + 1} + 1, \mathcal{H}\Big).$$

Since $\mathcal{E}$ contains an Expert parameterized for every $c \in \{0, ..., T - 1\}$, there always exists an expert $E_{\lceil \mathrm{M}_\mathcal{P}(x_{1:T}) \rceil} \in \mathcal{E}$ initialized with input $c = \lceil \mathrm{M}_\mathcal{P}(x_{1:T}) \rceil$. Running DWMA using these set of experts

---
**Algorithm 5** Online learner
---
**Input:** Hypothesis class $\mathcal{H}$, Offline learner $\mathcal{B}$, Time horizon $T$
1 For every $b \in \{0, ..., T-1\}$ let $E_b$ denote Algorithm 4 parameterized by input $b$.
2 Run the DWMA using $\{E_b\}_{b \in \{0,...,T-1\}}$ over the stream $(x_1, y_1), ..., (x_T, y_T)$.
---

$\mathcal{E}$ on the data stream $(x_1, y_1), ..., (x_T, y_T)$ ensures that our learner does not perform too much worse than $E_{\lceil M_{\mathcal{P}}(x_{1:T}) \rceil}$. Algorithm 5 formalizes this idea and Lemma 3.7 is proved in Appendix B.

**Lemma 3.7.** *For every $\mathcal{H} \subseteq \mathcal{Y}^{\mathcal{X}}$, Predictor $\mathcal{P}$, and no-regret offline learner $\mathcal{B}$, Algorithm 5 makes at most*

$$6\left( (M_{\mathcal{P}}(x_{1:T}) + 1)\overline{M}_{\mathcal{B}}\left( \frac{T}{M_{\mathcal{P}}(x_{1:T}) + 1} + 1, \mathcal{H} \right) + \log_2 T \right).$$

*mistakes in expectation on any realizable stream $(x_1, y_1), ..., (x_T, y_T)$.*

### 3.3 Lower bounds

In light of Theorem 3.1, it is natural ask whether the upper bounds derived in Section 3 are tight. A notable feature in upper bounds (ii) and (iii) is the product of the two mistake bounds $M_{\mathcal{P}}(x_{1:T})$ and $M_{\mathcal{B}}(T, \mathcal{H})$. Can this product can be replaced by a sum? Unfortunately, Theorem 3.8 shows that the upper bound in Theorem 3.1 can be tight.

**Theorem 3.8.** *Let $\mathcal{X} = [0, 1] \cup \{\star\}, \mathcal{Y} = \{0, 1\}$, and $\mathcal{H} = \{x \mapsto \mathbb{1}\{x \leq a\}\mathbb{1}\{x \neq \star\}\}$. Let $T, n \in \mathbb{N}$ be such that $n+1$ divides $T$ and $\frac{T}{n+1} + 1 = 2^k$ for some $k \in \mathbb{N}$. Then, there exists a Predictor $\mathcal{P}$ such that for every online learner $\mathcal{A}$ that uses $\mathcal{P}$ according to Protocol 2, there exists a realizable stream $(x_1, y_1), ..., (x_T, y_T)$ such that $M_{\mathcal{P}}(x_{1:T}) = n$ but*

$$\mathbb{E}\left[ \sum_{t=1}^{T} \mathbb{1}\{\mathcal{A}(x_t) \neq y_t\} \right] \geq \frac{(n+1)}{2} \log_2\left( \frac{T}{n+1} \right).$$

Theorem 3.8 shows that the upper bound in Theorem 3.1 is tight up to an additive factor in $\log_2 T$ because Lemma 2.1 gives that $\inf_{\mathcal{B}} \overline{M}_{\mathcal{B}}(T, \mathcal{H}) = O(\text{VC}(\mathcal{H}) \log_2 T)$ and $\text{VC}(\mathcal{H}) = 1$. The proof of Theorem 3.8 is technical and provided in Appendix E. Our proof involves four steps. First, we construct a class of streams $\mathcal{Z}_n \subseteq \mathcal{X}^{\star}$. Then, using $\mathcal{Z}_n$, we construct a deterministic, lazy, consistent Predictor $\mathcal{P}$ such that $\mathcal{P}$ makes mistakes exactly on timepoints $\{\frac{T}{n+1} + 1, ..., \frac{nT}{n+1} + 1\}$ for every stream $x_{1:T} \in \mathcal{Z}_n$. Next, whenever $x_{1:T} \in \mathcal{Z}_n$, we establish an equivalence between the game defined by Protocol 2 when given access to Predictor $\mathcal{P}$ and Online Classification with Peeks, a *different* game where there is no Predictor, but the learner observes the next $\frac{T}{n+1}$ examples at timepoints $t \in \{1, \frac{T}{n+1} + 1, ..., \frac{nT}{n+1} + 1\}$. Finally, for Online Classification with Peeks, we give a strategy for Nature such that it can force any online learner to make $\frac{(n+1)\log_2(\frac{T}{n+1})}{2}$ mistakes in expectation while ensuring that its selected stream satisfies $x_{1:T} \in \mathcal{Z}_n$ and $\inf_{h \in \mathcal{H}} \sum_{t=1}^{T} \mathbb{1}\{h(x_t) \neq y_t\} = 0$. A key component of the fourth step is the stream constructed by [Hanneke et al., 2024, Claim 3.4] to show that the minimax mistakes for classes with infinite Ldim is at least $\log_2 T$ in the offline setting.

**Remark.** Although Theorem 3.8 is stated using the class of one dimensional thresholds, it can be adapted to hold for any VC class with infinite Ldim as these classes embed thresholds [Alon et al., 2019, Theorem 3].

## 4 Discussion

In this paper, we initiated the study of online classification when the learner has access to machine-learned predictions about future examples. There are many interesting directions for future research and we list two below. Firstly, we only considered the classification setting, and it would be interested to extend our results to online scalar-valued regression. Secondly, we measure the performance of a Predictor through its mistake-bounds. When $\mathcal{X}$ is continuous, this might be an unrealistic measure of performance. Thus, it would be interesting to see whether our results can be generalized to the case where $\mathcal{X}$ is continuous and the guarantee of Predictors is defined in terms of $\ell_p$ losses.

## Acknowledgments and Disclosure of Funding

VR acknowledges the support from the NSF Graduate Research Fellowship Program.

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

# A Combinatorial dimensions

In this section, we review existing combinatorial dimensions in statistical learning theory. We start with the VC and Natarajan dimensions which characterize PAC learnability when $|\mathcal{Y}| = 2$ and $|\mathcal{Y}| < \infty$ respectively.

**Definition 3** (VC dimension). *A set $\{x_1, ..., x_n\} \in \mathcal{X}$ is shattered by $\mathcal{H}$, if $\forall y_1, ..., y_n \in \{0, 1\}$, $\exists h \in \mathcal{H}$, such that $\forall i \in [n]$, $h(x_i) = y_i$. The VC dimension of $\mathcal{H}$, denoted $\mathrm{VC}(\mathcal{H})$, is defined as the largest natural number $n \in \mathbb{N}$ such that there exists a set $\{x_1, ..., x_n\} \in \mathcal{X}$ that is shattered by $\mathcal{H}$.*

**Definition 4** (Natarajan Dimension). *A set $S = \{x_1, \ldots, x_d\}$ is shattered by a multiclass function class $\mathcal{H} \subseteq \mathcal{Y}^{\mathcal{X}}$ if there exist two witness functions $f, g : S \to \mathcal{Y}$ such that $f(x_i) \neq g(x_i)$ for all $i \in [d]$, and for every $\sigma \in \{0, 1\}^d$, there exists a function $h_\sigma \in \mathcal{H}$ such that for all $i \in [d]$, we have*

$$h_\sigma(x_i) = \begin{cases} f(x_i) & \text{if } \sigma_i = 1 \\ g(x_i) & \text{if } \sigma_i = 0 \end{cases} .$$

*The Natarajan dimension of $\mathcal{H}$, denoted $\mathrm{N}(\mathcal{H})$, is the size of the largest shattered set $S \subseteq \mathcal{X}$. If the size of the shattered set can be arbitrarily large, we say that $\mathrm{N}(\mathcal{H}) = \infty$.*

We note that $\mathrm{N}(\mathcal{H}) = \mathrm{VC}(\mathcal{H})$ whenever $|\mathcal{Y}| = 2$. Next, we move to the online setting, where the Littlestone dimension (Ldim) characterizes multiclass online learnability. To define the Ldim, we first define a Littlestone tree and a notion of shattering.

**Definition 5** (Littlestone tree). *A Littlestone tree of depth $d$ is a complete binary tree of depth $d$ where the internal nodes are labeled by examples of $\mathcal{X}$ and the left and right outgoing edges from each internal node are labeled by $0$ and $1$ respectively.*

Given a Littlestone tree $\mathcal{T}$ of depth $d$, a root-to-leaf path down $\mathcal{T}$ is a bitstring $\sigma \in \{0, 1\}^d$ indicating whether to go left ($\sigma_i = 0$) or to go right ($\sigma_i = 1$) at each depth $i \in [d]$. A path $\sigma \in \{0, 1\}^d$ down $\mathcal{T}$ gives a sequence of labeled examples $\{(x_i, \sigma_i)\}_{i=1}^d$, where $x_i$ is the example labeling the internal node following the prefix $(\sigma_1, ..., \sigma_{i-1})$ down the tree. A hypothesis $h_\sigma \in \mathcal{H}$ shatters a path $\sigma \in \{0, 1\}^d$, if for every $i \in [d]$, we have $h_\sigma(x_i) = \sigma_i$. In other words, $h_\sigma$ is consistent with the labeled examples when following $\sigma$. A Littlestone tree $\mathcal{T}$ is shattered by $\mathcal{H}$ if for every root-to-leaf path $\sigma$ down $\mathcal{T}$, there exists a hypothesis $h_\sigma \in \mathcal{H}$ that shatters it. Using this notion of shattering, we define the Littlestone dimension of a hypothesis class.

**Definition 6** (Littlestone dimension). *The Littlestone dimension of $\mathcal{H}$, denoted $\mathrm{L}(\mathcal{H})$, is the largest $d \in \mathbb{N}$ such that there exists a Littlestone tree $\mathcal{T}$ of depth $d$ shattered by $\mathcal{H}$. If there exists shattered Littlestone trees $\mathcal{T}$ of arbitrary depth, then we say that $\mathrm{L}(\mathcal{H}) = \infty$.*

Finally, the following notion of shattering is useful when proving the lower bound in Appendix E.

**Definition 7** (Threshold shattering). *A sequence $(x_1, ..., x_k) \in \mathcal{X}^k$ is threshold-shattered by $\mathcal{H} \subseteq \{0, 1\}^{\mathcal{X}}$ if there exists $(h_1, ..., h_k) \in \mathcal{H}^k$ such that $h_i(x_j) = \mathbb{1}\{j \leq i\}$ for all $i, j \in [k]$.*

# B Proof of Lemmas 3.6 and 3.7

*Proof.* (of Lemma 3.6) Let $(x_1, y_1), ..., (x_T, y_T)$ be the realizable stream to be observed by the Expert. For every $i \in \{0, ..., c\}$, let $m_i$ be the random variable denoting the number of mistakes made by $\mathcal{P}$ in rounds $\{\tilde{t}_i + 1, ..., \tilde{t}_{i+1}\}$. Recall that $\tilde{t}_0 = 0$ and $\tilde{t}_{c+1} = T$. Let $M = \sum_{i=0}^c m_i$ be the random variable denoting the total number of mistakes made by $\mathcal{P}$ on the realizable stream. Finally, let $\mathcal{A}$ denote Algorithm 4. Observe that,

$$\mathbb{E}\left[\sum_{t=1}^T \mathbb{1}\{\mathcal{A}(x_t) \neq y_t\}\right] = \mathbb{E}\left[\sum_{i=0}^c \sum_{t=\tilde{t}_i+1}^{\tilde{t}_{i+1}} \mathbb{1}\{\mathcal{A}(x_t) \neq y_t\}\right]$$

$$= \mathbb{E}\left[\sum_{i=0}^c \sum_{t=\tilde{t}_i+1}^{\tilde{t}_{i+1}} \mathbb{1}\{\mathcal{K}_i(x_t) \neq y_t\}\right]$$

$$\leq \mathbb{E}\left[\sum_{i=0}^c (m_i + 1)\overline{\mathrm{M}}_{\mathcal{B}}(\tilde{t}_{i+1} - \tilde{t}_i, \mathcal{H})\right]$$

where the first inequality follows from the guarantee of $\mathcal{K}$ and Lemma 2.2. Using Jensen's inequality, we get that

$$\mathbb{E}\left[\sum_{i=0}^{c}(m_i+1)\overline{\mathrm{M}}_{\mathcal{B}}(\tilde{t}_{i+1}-\tilde{t}_i,\mathcal{H})\right] \leq \mathbb{E}\left[\left(\sum_{i=0}^{c}(m_i+1)\right)\overline{\mathrm{M}}_{\mathcal{B}}\left(\frac{\sum_{i=0}^{c}(m_i+1)(\tilde{t}_{i+1}-\tilde{t}_i)}{\sum_{i=0}^{c}(m_i+1)},\mathcal{H}\right)\right]$$

$$= \mathbb{E}\left[\left(M+c+1\right)\overline{\mathrm{M}}_{\mathcal{B}}\left(\frac{\sum_{i=0}^{c}(m_i+1)(\tilde{t}_{i+1}-\tilde{t}_i)}{M+c+1},\mathcal{H}\right)\right]$$

$$= \mathbb{E}\left[\left(M+c+1\right)\overline{\mathrm{M}}_{\mathcal{B}}\left(\frac{\sum_{i=0}^{c}m_i(\tilde{t}_{i+1}-\tilde{t}_i)+T}{M+c+1},\mathcal{H}\right)\right]$$

$$= \mathbb{E}\left[\left(M+c+1\right)\overline{\mathrm{M}}_{\mathcal{B}}\left(\frac{\lceil\frac{T}{c+1}\rceil\sum_{i=0}^{c}m_i(i+1-i)+T}{M+c+1},\mathcal{H}\right)\right]$$

$$= \mathbb{E}\left[\left(M+c+1\right)\overline{\mathrm{M}}_{\mathcal{B}}\left(\frac{\lceil\frac{T}{c+1}\rceil M+T}{M+c+1},\mathcal{H}\right)\right].$$

Using the fact that $\lceil\frac{T}{c+1}\rceil \leq \frac{T}{c+1}+1$, we have

$$\mathbb{E}\left[\sum_{i=0}^{c}\sum_{j=0}^{m_i}\overline{\mathrm{M}}_{\mathcal{B}}(\tilde{t}_{i+1}-\tilde{t}_i,\mathcal{H})\right] \leq \mathbb{E}\left[\left(M+c+1\right)\overline{\mathrm{M}}_{\mathcal{B}}\left(\frac{\frac{MT}{c+1}+M+T}{M+c+1},\mathcal{H}\right)\right]$$

$$\leq \mathbb{E}\left[\left(M+c+1\right)\overline{\mathrm{M}}_{\mathcal{B}}\left(\frac{T}{c+1}+1,\mathcal{H}\right)\right]$$

$$= \left(\mathrm{M}_{\mathcal{P}}(x_{1:T})+c+1\right)\overline{\mathrm{M}}_{\mathcal{B}}\left(\frac{T}{c+1}+1,\mathcal{H}\right),$$

which completes the proof. $\qquad\square$

*Proof.* (of Lemma 3.7) Let $(x_1,y_1),...,(x_T,y_T)$ be the realizable stream to be observed by the learner. Let $\mathcal{A}$ denote the online learner in Algorithm 5. By the guarantees of the DWMA, we have

$$\mathbb{E}\left[\sum_{t=1}^{T}\mathbb{1}\{\mathcal{A}(x_t)\neq y_t\}\right] \leq 3\mathbb{E}\left[\inf_{b\in\{0,...,T-1\}}\sum_{t=1}^{T}\mathbb{1}\{E_b(x_t)\neq y_t\}\right]+3\log_2 T$$

$$\leq 3\mathbb{E}\left[\sum_{t=1}^{T}\mathbb{1}\{E_{\lceil\mathrm{M}_{\mathcal{P}}(x_{1:T})\rceil}(x_t)\neq y_t\}\right]+3\log_2 T$$

$$\leq 3(\mathrm{M}_{\mathcal{P}}(x_{1:T})+\lceil\mathrm{M}_{\mathcal{P}}(x_{1:T})\rceil+1)\overline{\mathrm{M}}_{\mathcal{B}}\left(\frac{T}{\lceil\mathrm{M}_{\mathcal{P}}(x_{1:T})\rceil+1}+1,\mathcal{H}\right)+3\log_2 T$$

$$\leq 6(\mathrm{M}_{\mathcal{P}}(x_{1:T})+1)\overline{\mathrm{M}}_{\mathcal{B}}\left(\frac{T}{\mathrm{M}_{\mathcal{P}}(x_{1:T})+1}+1,\mathcal{H}\right)+6\log_2 T,$$

where the last inequality follows from Lemma 3.6 and the fact that $\mathrm{M}_{\mathcal{P}}(x_{1:T}) \leq \lceil\mathrm{M}_{\mathcal{P}}(x_{1:T})\rceil \leq \mathrm{M}_{\mathcal{P}}(x_{1:T})+1$. $\qquad\square$

## C   Proof of Corollary 3.2 and 3.3

Using Theorem 3.1, we first show that for every $\mathcal{H}\subseteq\mathcal{Y}^{\mathcal{X}}$, Predictor $\mathcal{P}$, $\mathcal{Z}\subseteq\mathcal{X}^\star$, and no-regret offline learner $\mathcal{B}$, we have that

$$\inf_{\mathcal{A}}\mathrm{M}_{\mathcal{A}}(T,\mathcal{H},\mathcal{Z}) = O\left(\mathrm{L}(\mathcal{H})\wedge(\mathrm{M}_{\mathcal{P}}(T,\mathcal{Z})+1)\,\mathrm{M}_{\mathcal{B}}(T,\mathcal{H})\wedge\left((\mathrm{M}_{\mathcal{P}}(T,\mathcal{Z})+1)\,\overline{\mathrm{M}}_{\mathcal{B}}\left(\frac{T}{\mathrm{M}_{\mathcal{P}}(T,\mathcal{Z})+1},\mathcal{H}\right)+\log_2 T\right)\right).$$

*Proof.* It suffices to show that Algorithms 3 and 5 have mistake bounds $O((\mathrm{M}_{\mathcal{P}}(T, \mathcal{Z}) + 1) \mathrm{M}_{\mathcal{B}}(T, \mathcal{H}))$ and $O\left((\mathrm{M}_{\mathcal{P}}(T, \mathcal{Z}) + 1) \overline{\mathrm{M}}_{\mathcal{B}}\left(\frac{T}{\mathrm{M}_{\mathcal{P}}(T, \mathcal{Z})+1}, \mathcal{H}\right) + \log_2 T\right)$ respectively. To see that Algorithm 3's mistake bounds is $O((\mathrm{M}_{\mathcal{P}}(T, \mathcal{Z}) + 1) \mathrm{M}_{\mathcal{B}}(T, \mathcal{H}))$, note that $\mathrm{M}_{\mathcal{P}}(x_{1:T}) \leq \mathrm{M}_{\mathcal{P}}(T, \mathcal{Z})$ for every $x_{1:T} \in \mathcal{Z}$. To see that Algorithm 5's expected mistake bound is $O\left((\mathrm{M}_{\mathcal{P}}(T, \mathcal{Z}) + 1) \overline{\mathrm{M}}_{\mathcal{B}}\left(\frac{T}{\mathrm{M}_{\mathcal{P}}(T, \mathcal{Z})+1}, \mathcal{H}\right) + \log_2 T\right)$, we follow the exact same proof strategy as in the proof of Lemma 3.7, but picking a different expert when upper bounding the expected number of mistakes. Namely, following the same steps as in the proof of Lemma 3.7, we have that

$$\mathbb{E}\left[\sum_{t=1}^{T} \mathbb{1}\{\mathcal{A}(x_t) \neq y_t\}\right] \leq 3\mathbb{E}\left[\inf_{b \in \{0,\dots,T-1\}} \sum_{t=1}^{T} \mathbb{1}\{E_b(x_t) \neq y_t\}\right] + 3\log_2 T$$

where $\mathcal{A}$ denotes Algorithm 5. Picking $b = \lceil \mathrm{M}_{\mathcal{P}}(T, \mathcal{Z}) \rceil$, using Lemma 3.6, and the fact that $\mathrm{M}_{\mathcal{P}}(x_{1:T}) \leq \mathrm{M}_{\mathcal{P}}(T, \mathcal{Z})$ gives the desired upper bound on $\mathbb{E}\left[\sum_{t=1}^{T} \mathbb{1}\{\mathcal{A}(x_t) \neq y_t\}\right]$ of

$$O\left((\mathrm{M}_{\mathcal{P}}(T, \mathcal{Z}) + 1)\overline{\mathrm{M}}_{\mathcal{B}}\left(\frac{T}{\mathrm{M}_{\mathcal{P}}(T, \mathcal{Z}) + 1}, \mathcal{H}\right) + \log_2 T\right), \tag{1}$$

completing the proof. $\qquad\square$

Corollary 3.2 follows from the fact that the upper bound on $\inf_{\mathcal{A}} \mathrm{M}_{\mathcal{A}}(T, \mathcal{H}, \mathcal{Z})$ is sublinear whenever $\mathrm{M}_{\mathcal{P}}(T, \mathcal{Z}) = o(T)$ and $\mathrm{M}_{\mathcal{B}}(T, \mathcal{H}) = o(T)$. To get Corollary 3.3, recall that by Lemma 2.1, there exists an offline learner $\mathcal{B}$ such that

$$\overline{\mathrm{M}}_{\mathcal{B}}(T, \mathcal{H}) = O\left(\mathrm{VC}(\mathcal{H}) \log_2 T\right).$$

Plugging this bound into upper bound (1) completes the proof.

## D   Proof of Theorem 3.1

Let $\mathcal{A}$ denote the DWMA using the Standard Optimal Algorithm (SOA), Algorithm 3 and Algorithm 5 as experts. Then, for any realizable stream $(x_1, y_1), \dots, (x_T, y_T)$, Lemma 3.4 gives that

$$\mathbb{E}\left[\sum_{t=1}^{T} \mathbb{1}\{\mathcal{A}(x_t) \neq y_t\}\right] \leq 3\mathbb{E}\left[\min_{i \in [3]} M_i + \log_2 3\right] \leq 3\min_{i \in [3]} \mathbb{E}\left[M_i\right] + 5,$$

where we take $M_1, M_2$ and $M_3$ to be the number of mistakes made by the SOA, Algorithm 3, and Algorithm 5 respectively. Note that $M_2$ and $M_3$ are random variables since $\mathcal{B}$ and $\mathcal{P}$ may be randomized algorithms. Finally, using Lemma 3.5, Lemma 3.7 and the fact that the SOA makes at most $\mathrm{L}(\mathcal{H})$ mistakes on any realizable stream [Littlestone, 1987] completes the proof of Theorem 3.1.

## E   Proof of Theorem 3.8

Let $\mathcal{X} = \mathbb{R} \cup \{\star\}$ and $\mathcal{H} = \{x \mapsto \mathbb{1}\{x \leq a\} \mathbb{1}\{x \neq \star\}\}$. Let $T, n \in \mathbb{N}$ be chosen such that $T$ is a multiple of $n+1$ and $\frac{T}{n+1} + 1 = 2^k$ for some $k \in \mathbb{N}$. Our proof of Theorem 3.8 will be in four steps, as described below.

(1) We construct a class of streams $\mathcal{Z}_n \subseteq \mathcal{X}^\star$.

(2) Using $\mathcal{Z}_n$, we construct a deterministic, lazy, consistent Predictor $\mathcal{P}$ such that $\mathcal{P}$ makes mistakes exactly on timepoints $\{\frac{T}{n+1} + 1, \dots, \frac{nT}{n+1} + 1\}$ for every stream $x_{1:T} \in \mathcal{Z}_n$.

(3) When $x_{1:T} \in \mathcal{Z}_n$, we establish an equivalence between the game defined by Protocol 2 when given access to Predictor $\mathcal{P}$ and Online Classification with Peeks, a *different* game where there is no Predictor, but the learner observes the next $\frac{T}{n+1}$ examples at timepoints $t \in \{1, \frac{T}{n+1} + 1, ..., \frac{nT}{n+1} + 1\}$.

(4) For Online Classification with Peeks, we give a strategy for Nature such that it can force any online learner to make $\frac{(n+1)\log_2(\frac{T}{n+1})}{2}$ mistakes in expectation while ensuring that the stream of labeled examples it picks $(x_1, y_1), ..., (x_T, y_T)$ satisfies the constraint that $x_{1:T} \in \mathcal{Z}_n$ and $\inf_{h \in \mathcal{H}} \sum_{t=1}^{T} \mathbb{1}\{h(x_t) \neq y_t\} = 0$.

Composing steps 1-4 shows the existence of a Predictor $\mathcal{P}$ such that for any learner $\mathcal{A}$ playing Protocol 2 using $\mathcal{P}$, there exists a realizable stream where $\mathcal{A}$ makes at least $\frac{(n+1)}{2}\log_2(\frac{T}{n+1})$ mistakes in expectation.

**Step 1: Construction of $\mathcal{Z}_n$**

Let $\mathcal{S}$ be the set of all *strictly* increasing sequences of real numbers in $(0, 1)$ of size $\frac{T}{n+1}$. Fix a function $f : \mathbb{R}^2 \to \mathcal{S}$ which, given $a < b \in \mathbb{R}$, outputs an element of $\mathcal{S}$ that lies strictly in between $a$ and $b$. For example, given $a < b \in \mathbb{R}$, the function $f$ can output evenly spaced real numbers of size $\frac{T}{n+1}$. Let $\mathrm{Dyd} : \mathcal{S} \to \mathcal{X}^{\frac{T}{n+1}}$ be a function that reorders the input $S \in \mathcal{S}$ in Dyadic order. Namely, if $S = (x_1, ..., x_N)$ where $N + 1 = 2^k$ for some $k \in \mathbb{N}$, then $\mathrm{Dyd}(S)$ is

$$x_{\frac{N}{2}}, x_{\frac{N}{4}}, x_{\frac{3N}{4}}, x_{\frac{N}{8}}, x_{\frac{3N}{8}}, x_{\frac{5N}{8}}, x_{\frac{7N}{8}}, ..., x_{\frac{(2^k-1)N}{2^k}}.$$

See the Proof of Claim 3.4 in Hanneke et al. [2024] for a more detailed description of a Dyadic order. On the other hand, let $\mathrm{Sort} : \mathcal{X}^{\frac{T}{n+1}} \to \mathcal{S}$ be a function that reorders its input in increasing order. Let $\mathcal{J} := \{1, ..., \frac{T}{n+1} + 1\}^{\leq n}$ be the set of all sequences of indices of length at most $n$ taking values in $\{1, ..., \frac{T}{n+1} + 1\}$. For the remainder of this section, we will use $S_i$ to denote the $i$th element in a sequence $S \in \mathcal{S}$. Moreover, for any two sequences $S^1, S^2 \in \mathcal{S}$, we say $S^1 < S^2$ if $S^1_{|S^1|} < S^2_1$. That is, $S^1 < S^2$, if $S^1$ lies strictly to the left of $S^2$.

---

**Algorithm 6** Stream Generator (SG)

---

**Input:** $S^0 \in \mathcal{S}, j_{1:m} \in \mathcal{J}$
1 **Initialize:** $a_0 = 0, b_0 = 1$
2 **for** $i = 1, ..., m$ **do**
3     **if** $j_i = 1$ **then**
4         $S^i \leftarrow f(a_{i-1}, S^{i-1}_1)$
5         $a_i \leftarrow a_{i-1}$
6         $b_i \leftarrow S^{i-1}_1$
7     **else if** $j_i = \frac{T}{n+1} + 1$ **then**
8         $S_i \leftarrow f(S^{i-1}_{\frac{T}{n+1}}, b_{i-1})$
9         $a_i \leftarrow S^{i-1}_{\frac{T}{n+1}}$
10         $b_i \leftarrow b_{i-1}$
11     **else**
12         $S_i \leftarrow f(S^{i-1}_{j-1}, S^{i-1}_j)$
13         $a_i \leftarrow S^{i-1}_{j-1}$
14         $b_i \leftarrow S^{i-1}_j$
15     **end**
16 **end**
17 **Return:** $\mathrm{Dyd}(S^0) \circ ... \circ \mathrm{Dyd}(S^m)$

---

We will construct a stream for every sequence $j_{1:m} \in \mathcal{J}, m \leq n$, algorithmically as follows. Fix $S^0 := f(0, 1) \in \mathcal{S}$ and let SG denote Algorithm 6. Let

$$\mathcal{Z}_n = \Big\{\mathrm{SG}(S^0, j_{1:m}) : j_{1:m} \in \mathcal{J}, m \in \{1, ..., n\}\Big\}$$

denote the stream class generated by applying SG to inputs $S^0$ and $j_{1:m}$ for every $j_{1:m} \in \mathcal{J}, m \leq n$. We make four important observations about $\mathcal{Z}_n$, which we will use to construct a Predictor that can reconstruct $S^i$ given $S^0$ and the first example of the block $S^{i-1}$.

**Observation 1.** *For every sequence $x_{1:T} \in \mathcal{Z}$, we have that $x_{1:\frac{T}{n+1}} = \mathrm{Dyd}(S^0)$.*

The first observation follows from the fact that the same initial sequence $S^0$ is used to generate every stream in $\mathcal{Z}_n$.

**Observation 2.** *For any pair $j_{1:n}^1, j_{1:n}^2 \in \mathcal{J}$ and $m \leq n$, if $j_{1:m}^1 = j_{1:m}^2$, then $\mathrm{SG}(S^0, j_{1:m}^1) = \mathrm{SG}(S^0, j_{1:m}^2)$.*

The second observation follows from the fact that SG is deterministic.

**Observation 3.** *For every $x_{1:T} \in \mathcal{Z}_n$ such that $x_{1:T} := \mathrm{SG}(S^0, j_{1:n}) = \mathrm{Dyd}(S^0) \circ ... \circ \mathrm{Dyd}(S^n)$, the index $j_i$ can be computed exactly using only $S^{i-1}$ and $S_1^i$ for every $i \in [n]$.*

To see the third observation, fix some $x_{1:T} \in \mathcal{Z}_n$. Then, there exists a sequence $S^1, ..., S^n \in \mathcal{S}$ such that $x_{1:T} = \mathrm{Dyd}(S^0) \circ ... \circ \mathrm{Dyd}(S^n)$ as well as a sequence $(a_0, b_0), ..., (a_n, b_n)$. In addition, there exists a $j_{1:n} \in \mathcal{J}$ such that $x_{1:T} = \mathrm{SG}(S^0, j_{1:n})$. Fix $i \in [n]$ and consider $S^{i-1}$ and $S^i$. By definition of Algorithm 6, there exists an index $q \in \{1, ..., \frac{T}{n+1} + 1\}$ such that $S^i = f(S_{q-1}^{i-1}, S_q^{i-1})$ where we take $S_0^{i-1} = a_{i-1}$ and $S_{\frac{T}{n+1}+1}^{i-1} = b_{i-1}$. We claim that the index $q$ is unique. This follows from the fact that the collection $\{f(S_j^{i-1}, S_{j+1}^{i-1})\}_{j=0}^{\frac{T}{n+1}}$ is pairwise disjoint since $a_{i-1} = S_0^{i-1} < S_1^{i-1} < ... < S_{\frac{T}{n+1}}^{i-1} < S_{\frac{T}{n+1}+1}^{i-1} = b_{i-1}$. Finally, we claim that $S^{i-1}$ and the element $S_1^i$ identifies the index $q$. This follows because $f(a_{i-1}, S_1^{i-1}) < f(S_1^{i-1}, S_2^{i-1}) < ... < f(S_{\frac{T}{n+1}-1}^{i-1}, S_{\frac{T}{n+1}}^{i-1}) < f(S_{\frac{T}{n+1}}^{i-1}, b_{i-1})$ and thus $q$ is the smallest index $p \in \{1, ..., \frac{T}{n+1}\}$ such that $S_1^i < S_p^{i-1}$ and $\frac{T}{n+1} + 1$ if such a $p$ does not exist.

**Observation 4.** *Fix a sequence $j_{1:n} \in \mathcal{J}$ and let $\mathrm{Dyd}(S^0) \circ ... \circ \mathrm{Dyd}(S^n) = \mathrm{SG}(S^0, j_{1:n})$. For every $i, p \in [n]$ such that $i < p$, we have that:*

*(i)* $S^p < S^i$ *if* $j_i = 1$;

*(ii)* $S_{j_i-1}^i < S^p < S_{j_i}^i$ *if* $2 \leq j_i \leq \frac{T}{n+1}$;

*(iii)* $S^i < S^p$ *if* $j_i = \frac{T}{n+1} + 1$.

The fourth observation follows from the fact that for every $i \in [n]$ and index $j_i \in \{1, ..., \frac{T}{n+1} + 1\}$, the remaining sequence of sets $S^{i+1}, ..., S^n$ all lie in the interval $(S_{j_i-1}^i, S_{j_i}^i)$ by design of Algorithm 6, where again we take $S_0^{i-1} = a_{i-1} < S_1^{i-1}$ and $S_{\frac{T}{n+1}+1}^{i-1} = b_{i-1} > S_{\frac{T}{n+1}}^{i-1}$.

**Step 2: Constructing a Predictor for $\mathcal{Z}_n$**

We now show that Algorithm 7 is a lazy, consistent Predictor for $\mathcal{Z}_n$ that only makes mistakes at timepoints $\{\frac{T}{n+1} + 1, ..., \frac{nT}{n+1} + 1\}$.

**Lemma E.1.** *For any sequence $x_{1:T} \in \mathcal{Z}_n$, Algorithm 7 is a lazy, consistent Predictor for $\mathcal{Z}_n$ that only makes mistakes at timepoints $\{\frac{T}{n+1} + 1, ..., \frac{nT}{n+1} + 1\}$.*

*Proof.* Let $\mathcal{P}$ denote Algorithm 7 and $x_{1:T} \in \mathcal{Z}_n$. Then, there exists $S^1, ..., S^n \in \mathcal{S}$ and a sequence of indices $j_{1:n} \in \mathcal{J}$ such that $x_{1:T} = \mathrm{Dyd}(S^0) \circ \mathrm{Dyd}(S^1) \circ ... \circ \mathrm{Dyd}(S^n) = \mathrm{SG}(S^0, j_{1:n})$.

We now prove that $\mathcal{P}$ makes mistakes only on timepoints $\{\frac{T}{n+1} + 1, ..., \frac{nT}{n+1} + 1\}$ and no where else. Our proof is by induction using the following inductive hypothesis. For every $i \in \{1, ..., n\}$, we have that $\mathcal{P}$:

*(i)* sets $J_{1:i} = j_{1:i}$ on round $\frac{iT}{n+1} + 1$;

*(ii)* makes mistakes on rounds $\{\frac{T}{n+1} + 1, \frac{2T}{n+1} + 1, ..., \frac{iT}{n+1} + 1\}$ and no where in between.

---

**Algorithm 7** Predictor for $\mathcal{Z}_n$

---

**Input:** $\mathcal{Z}_n$

1 **Initialize:** $J = ()$
2 **for** $t = 1, .., T$ **do**
3 $\quad$ Receive $x_t$
4 $\quad$ **if** $t = 1$ **then**
5 $\quad\quad$ Set $\hat{x}^t_{1:T} = \mathrm{Dyd}(S^0) \circ \hat{x}_{\frac{T}{n+1}+1:T}$ where $\hat{x}_{\frac{T}{n+1}+1:T} = (\star, ..., \star)$.
6 $\quad$ **else if** $t = \frac{iT}{n+1} + 1\ for\ some\ i \in \{1, ..., n\}$ **then**
7 $\quad\quad$ Let $S = \mathrm{Sort}(x_{t-\frac{T}{n+1}:t-1})$ be the last $\frac{T}{n+1}$ examples sorted in increasing order.
8 $\quad\quad$ Find the smallest $j \in \{1, ..., \frac{T}{n+1}\}$ such that $x_t < S_j$. If no such $j$ exists, set $j = \frac{T}{n+1} + 1$.
9 $\quad\quad$ Update $J \leftarrow J \circ j$.
10 $\quad\quad$ Set $\hat{x}^t_{1:T} = \mathrm{SG}(S^0, J) \circ \hat{x}_{t+\frac{T}{n+1}:T}$ where $\hat{x}_{t+\frac{T}{n+1}:T} = (\star, ..., \star)$.
11 $\quad$ **else**
12 $\quad\quad$ Set $\hat{x}^t_{1:T} \leftarrow \hat{x}^{t-1}_{1:T}$.
13 $\quad$ **end**
14 $\quad$ Predict $\hat{x}^t_{1:T}$.
15 **end**

---

For the base case, let $i = 1$. $\mathcal{P}$ does not make any mistakes in $\{1, 2, ..., \frac{T}{n+1}\}$ since it knows $S^0$ using $\mathcal{Z}_n$, computes $x_{1:\frac{T}{n+1}} = \mathrm{Dyd}(S^0)$ in line 5, and does not change its prediction until round $\frac{iT}{n+1} + 1$ based on line 6. At time point $t_1 = \frac{T}{n+1} + 1$, $\mathcal{P}$ makes a mistake since $\hat{x}^{t_1-1}_{t_1} = \star \neq x_{t_1}$. Moreover, using Observation 3, the index $j \in \{1, ..., \frac{T}{n+1}\}$ computed in round $t_1$ on line 8 matches $j_1$. Thus, we have that $J_1 = j_1$. This completes the base case.

Now for the induction step, let $i \in \{2, ..., n\}$. Suppose that the induction step is true for $i - 1$. This means that $\mathcal{P}$:

(i) sets $J_{1:i-1} = j_{1:i-1}$ on round $\frac{(i-1)T}{n+1} + 1$;

(ii) makes mistakes on rounds $\{\frac{T}{n+1} + 1, \frac{2T}{n+1} + 1, ..., \frac{(i-1)T}{n+1} + 1\}$ and no where in between.

We need to show that $\mathcal{P}$ sets $J_i = j_i$ on round $\frac{iT}{n+1} + 1$, $\mathcal{P}$ makes no mistakes between $\frac{(i-1)T}{n+1} + 2$ and $\frac{iT}{n+1}$, but makes a mistake at $\frac{iT}{n+1} + 1$. At timepoint $t_{i-1} = \frac{(i-1)T}{n+1} + 1$, $\mathcal{P}$ computes $J_{i-1} = j_{i-1}$ (by assumption) and thus sets $\hat{x}^{t_{i-1}}_{1:T} = \mathrm{SG}(S^0, J_{1:i-1}) = \mathrm{SG}(S^0, (j_1, ..., j_{i-1})) = \mathrm{Dyd}(S^0) \circ ... \circ \mathrm{Dyd}(S^{i-1})$ using Observation 2. Therefore, $\mathcal{P}$ predicts on round $t_{i-1}$ the sequence $\hat{x}^{t_{i-1}}_{1:T} = x_{1:\frac{iT}{n+1}} \circ (\star, ..., \star)$, implying that $\mathcal{P}$ makes no mistakes for rounds $\frac{(i-1)T}{n+1} + 2, ..., \frac{iT}{n+1}$ since it does not change its prediction until round $\frac{iT}{n+1} + 1$ by line 12. However, since $\hat{x}^{t_i-1}_{t_i} = \star$, $\mathcal{P}$ makes a mistake on round $t_i = \frac{iT}{n+1} + 1$. Finally, by Observation 3, the example $x_{t_i}$ and the previously observed sequence $x_{t_{i-1}:t_i-1}$ gives away $j_i$, thus $\mathcal{P}$ sets $J_i = j_i$ on line 8 in round $t = \frac{iT}{n+1} + 1$. This completes the induction step and the proof of the claim that $\mathcal{P}$ only makes mistakes on timepoints $\{\frac{T}{n+1} + 1, ..., \frac{nT}{n+1} + 1\}$. To see that $\mathcal{P}$ is lazy, observe that by line 12, $\mathcal{P}$ does not update its prediction on rounds in between those in $\{\frac{T}{n+1} + 1, ..., \frac{nT}{n+1} + 1\}$. To see that $\mathcal{P}$ is consistent, note that $\mathcal{P}$ uses prefixes of $j_1, ..., j_n$, $S^0$, and SG to compute its predictions in line 10. Thus, consistency follows from Observation 2. $\qquad\square$

**Step 3: Equivalence to Online Classification with Peeks**

For any stream $x_{1:T} \in \mathcal{Z}_n$, having access to the Predictor specified by Algorithm 7 implies that at every $t \in \{1, \frac{T}{n+1} + 1, ..., \frac{nT}{n+1} + 1\}$, the learner observes predictions $\hat{x}^t_{1:T}$ where $\hat{x}^t_{1:t-1} = x_{1:t-1}$, $\hat{x}^t_{t:t+\frac{T}{n+1}} = x_{t:t+\frac{T}{n+1}}$, and $\hat{x}^t_{t+\frac{T}{n+1}+1:T} = (\star, ..., \star)$. Accordingly, at the timepoints $t \in \{1, \frac{T}{n+1} + 1, ..., \frac{nT}{n+1} + 1\}$, the learner observes the next $\frac{T}{n+1} - 1$ examples $x_{t:t+\frac{T}{n+1}}$ in the stream, but learns

nothing about the future examples $x_{t+\frac{T}{n+1}+1:T}$. In addition, for timepoints in between those in $\{1, \frac{T}{n+1}+1, ..., \frac{nT}{n+1}+1\}$, the learner does not observe any new information from $\mathcal{P}$ since by line 12 in Algorithm 7, $\hat{x}^i_{1:T} = \hat{x}^{i+r}_{1:T}$ for every $i \in \{1, \frac{T}{n+1}+1, ..., \frac{nT}{n+1}+1\}$ and $r \in \{1, ..., \frac{T}{n+1}-1\}$. As a result, whenever $x_{1:T} \in \mathcal{Z}_n$, Protocol 2 with the Predictor specified by Algorithm 7 is equivalent to the setting we call Online Classification with Peeks where there is no Predictor, but the learner observes the next $\frac{T}{n+1}-1$ examples exactly at timepoints $t \in \{1, \frac{T}{n+1}+1, ..., \frac{nT}{n+1}+1\}$. Indeed, by having knowledge of the next $\frac{T}{n+1}-1$ examples exactly at timepoints $t \in \{1, \frac{T}{n+1}+1, ..., \frac{nT}{n+1}+1\}$, a learner for Online Classification with Peeks can simulate a Predictor that acts like Algorithm 7. Likewise, a learner for Online Classification with Predictions can use Algorithm 7 to simulate an adversary that reveals the next $\frac{T}{n+1}-1$ examples exactly at timepoints $t \in \{1, \frac{T}{n+1}+1, ..., \frac{nT}{n+1}+1\}$. Accordingly, we consider Online Classification with Peeks for the rest of the proof and show how Nature can force the lower bound in Theorem 3.8 under this new setting.

**Step 4: Nature's Strategy for Online Classification with Peeks**

Let $\mathcal{A}$ be any online learner and consider the game where the learner $\mathcal{A}$ observes the next $\frac{T}{n+1}-1$ examples at timepoints $\{1, \frac{T}{n+1}+1, ..., \frac{nT}{n+1}+1\}$. We construct a hard stream for $\mathcal{A}$ in this setting. We first describe a minimax optimal *offline* strategy for Nature when it is forced to play a sequence of examples $S \in \mathcal{S}$ sorted in Dyadic order.

---

**Algorithm 8** Nature's Minimax Offline Strategy

---

**Input:** $\tilde{S} = \mathrm{Dyd}(S)$ for some $S \in \mathcal{S}$, Version space $V \subseteq \{0,1\}^{\mathcal{X}}$
**Initialize:** $V_1 = V$
Reveal $\tilde{S}$ to the learner $\mathcal{A}$.
**for** $t = 1, ..., \frac{T}{n+1}$ **do**
    Observe the probability $\hat{p}_t$ of $\mathcal{A}$ predicting label 1.
    **if** $\hat{p}_t \geq 1/2$ **then**
        If there exists $h \in V_t$ such that $h(x_t) = 0$, reveal true label $y_t = 0$. Else, reveal $y_t = 1$.
    **else**
        If there exists $h \in V_t$ such that $h(x_t) = 1$, reveal true label $y_t = 1$. Else, reveal $y_t = 0$.
    Update $V_{t+1} = \{h \in V_t : h(x_t) = y_t\}$.
**end**
**Return:** True labels $y_1, ... y_{\frac{T}{n+1}}$, Version space $V_{\frac{T}{n+1}+1}$

---

**Lemma E.2.** *For any learner $\mathcal{A}$, $\tilde{S} = \mathrm{Dyd}(S)$, and Version space $V \subseteq \{0,1\}^{\mathcal{X}}$, Algorithm 8 forces $\mathcal{A}$ to make at least $\frac{1}{2}\log_2(\frac{T}{n+1})$ mistakes in expectation if $S$ is threshold-shattered (Definition 7) by $V$.*

*Proof.* The lemma follows directly from Theorem 3.4 in Hanneke et al. [2024]. $\qquad\square$

For the definition of threshold-shattering, see Appendix A. Note that for every input $\tilde{S} = \mathrm{Dyd}(S)$ and $V \subseteq \{0,1\}^{\mathcal{X}}$ to Algorithm 8, its output version space $V_{|\tilde{S}|+1}$ is non-empty and consistent with the sequence $(\tilde{S}_1, y_1), ..., (\tilde{S}_{\frac{T}{n+1}}, y_{\frac{T}{n+1}})$ as long as $|V| > 0$. This property will be crucial when proving Lemma E.4. We are now ready to describe Nature's strategy for Online Classification with Peeks. The pseudocode is provided in Algorithm 9.

We establish a series of important lemmas.

**Lemma E.3.** *For every learner $\mathcal{A}$, if $(x_1, y_1), ..., (x_T, y_T)$ is the stream the output of Algorithm 9 when playing against $\mathcal{A}$, then $x_{1:T} \in \mathcal{Z}_n$.*

*Proof.* Fix a learner $\mathcal{A}$ and let $(x_1, y_1), ..., (x_T, y_T)$ denote the output of Algorithm 9 playing against $\mathcal{A}$. Let $j_{1:n}$ denote the sequences of indices output by Algorithm 9. Then, since SG is deterministic, by line 6-7 in Algorithm 9, we have that $x_{1:T} = \mathrm{SG}(S^0, (j_1, ..., j_n)) \in \mathcal{Z}_n$. $\qquad\square$

**Lemma E.4.** *For every learner $\mathcal{A}$, if $(x_1, y_1), ..., (x_T, y_T)$ is the stream the output of Algorithm 9 when playing against $\mathcal{A}$, then $(x_1, y_1), ..., (x_T, y_T)$ is realizable by $\mathcal{H}$.*

---

**Algorithm 9** Nature's Strategy for Online Classification with Peeks

---

**Input:** Learner $\mathcal{A}$, Hypothesis class $\mathcal{H}$

1  **Initialize:** $V_1 = \mathcal{H}$
2  **for** $i = 1, .., n+1$ **do**
3      **if** $i = 1$ **then**
4          Set $x_{1:\frac{T}{n+1}} = \mathrm{Dyd}(S^0)$ and reveal it to the learner $\mathcal{A}$.
5      **else**
6          Compute $S = \mathrm{SG}(S^0, (j_1, ..., j_{i-1}))$.
7          Let $x_{\frac{(i-1)T}{n+1}+1:\frac{iT}{n+1}}$ be the last $\frac{T}{n+1}$ examples in $S$ and reveal it to the learner $\mathcal{A}$.
8      Play against $\mathcal{A}$ according to Algorithm 8 using $x_{\frac{(i-1)T}{n+1}+1:\frac{iT}{n+1}}$ and version space $V_i$.
9      Let $y_{\frac{(i-1)T}{n+1}+1}, ..., y_{\frac{iT}{n+1}}$ be the returned labels and $V_{i+1} \subseteq V_i$ be the returned version space.
10     Let $\tilde{y}_{\frac{(i-1)T}{n+1}+1}, ..., \tilde{y}_{\frac{iT}{n+1}}$ be the sequence of true labels after sorting

$$\left(x_{\frac{(i-1)T}{n+1}+1}, y_{\frac{(i-1)T}{n+1}+1}\right), ..., \left(x_{\frac{iT}{n+1}}, y_{\frac{iT}{n+1}}\right)$$

11     in increasing order with respect to the examples.
12     **if** $\tilde{y}_{\frac{iT}{n+1}} = 1$ **then**
13         Set $j_i = \frac{T}{n+1} + 1$.
14     **else**
15         Set $j_i$ to be the smallest $p \in \{1, ..., \frac{T}{n+1}\}$ such that $\tilde{y}_{\frac{(i-1)T}{n+1}+p} = 0$.
16 **end**
17 **Return:** Stream $(x_1, y_1), ..., (x_T, y_T)$, indices $j_{1:n}$, and version spaces $V_1, ..., V_{n+2}$.

---

*Proof.* Fix a learner $\mathcal{A}$ and let $(x_1, y_1), ..., (x_T, y_T)$ be the output of Algorithm 9 when playing against $\mathcal{A}$. Let $V_2, ..., V_{n+2}$ be the sequence of version spaces output by Algorithm 9. It suffices to show that $V_{n+2}$ is not empty and is consistent with $(x_1, y_1), ..., (x_T, y_T)$. Our proof will be by induction using the following hypothesis: $V_{i+1}$ is non-empty and consistent with the sequence $(x_1, y_1), ..., (x_{\frac{iT}{n+1}}, y_{\frac{iT}{n+1}})$. For the base case, let $i = 1$. Then, by Algorithm 8, line 8 in Algorithm 9, and the fact that $|V_1| = |\mathcal{H}| > 0$, we have that $|V_2| > 0$ and $V_2$ is consistent with $(x_1, y_1), ..., (x_{\frac{T}{n+1}}, y_{\frac{T}{n+1}})$. Now consider some $i \geq 2$ and suppose the induction hypothesis is true for $i - 1$, Then, we know that $|V_i| > 0$ and $V_i$ is consistent with $(x_1, y_1), ..., (x_{\frac{(i-1)T}{n+1}}, y_{\frac{(i-1)T}{n+1}})$. Again, by design of Algorithm 8 and line 9 in Algorithm 9, it follows that $|V_{i+1}| > 0$ and $V_{i+1}$ is consistent with $(x_{\frac{(i-1)T}{n+1}+1}, y_{\frac{(i-1)T}{n+1}+1}), ..., (x_{\frac{iT}{n+1}}, y_{\frac{iT}{n+1}})$. Since $V_{i+1} \subseteq V_i$, and $V_i$ is consistent with $(x_1, y_1), ..., (x_{\frac{(i-1)T}{n+1}}, y_{\frac{(i-1)T}{n+1}})$, we get that $V_{i+1}$ is consistent with $(x_1, y_1), ..., (x_{\frac{iT}{n+1}}, y_{\frac{iT}{n+1}})$, completing the induction step. $\qquad\square$

**Lemma E.5.** *For every learner $\mathcal{A}$, if $(x_1, y_1), ..., (x_T, y_T)$ and $V_1, ..., V_{n+2}$ are stream and version spaces output by Algorithm 9 when playing against $\mathcal{A}$, then for every $i \in \{1, ..., n+1\}$, the version space $V_i$ threshold-shatters $x_{\frac{(i-1)T}{n+1}+1:\frac{iT}{n+1}}$.*

*Proof.* Fix a learner $\mathcal{A}$ and let $(x_1, y_1), ..., (x_T, y_T)$ denote the output of Algorithm 9 playing against $\mathcal{A}$. Let $j_{1:n}$ and $V_1, ..., V_{n+2}$ denote the sequences of indices and version spaces output by Algorithm 9 respectively. Note that $x_{1:T} = \mathrm{SG}(S^0, j_{1:n})$. Moreover, for every $i \in \{2, ..., n\}$, we have that $x_{1:\frac{iT}{n+1}} = \mathrm{SG}(S^0, (j_1, ..., j_{i-1}))$ by lines 6-7.

Fix an $i \in \{1, ..., n+1\}$. It suffices to show that the hypotheses parameterized by $x_{\frac{(i-1)T}{n+1}+1:\frac{iT}{n+1}}$ belong in $V_i$. Our proof will be by induction. For the base case, since $V_1 = \mathcal{H}$, it trivially follows that the hypothesis parameterized by $x_{\frac{(i-1)T}{n+1}+1:\frac{iT}{n+1}}$ belong to $V_1$. Now, suppose that $x_{\frac{(i-1)T}{n+1}+1:\frac{iT}{n+1}}$ belong to $V_m$ for some $m < i$. We show that $V_{m+1}$ also contains the hypothesis parameterized by $x_{\frac{(i-1)T}{n+1}+1:\frac{iT}{n+1}}$. Recall that $V_{m+1} \subseteq V_m$ is the subset of $V_m$ that is consistent with the labeled data

$$\left(x_{\frac{(m-1)T}{n+1}+1}, y_{\frac{(m-1)T}{n+1}+1}\right), ..., \left(x_{\frac{mT}{n+1}}, y_{\frac{mT}{n+1}}\right)$$

and is the result of running Algorithm 8 with input version space $V_m$ and sequence $x_{\frac{(m-1)T}{n+1}+1:\frac{mT}{n+1}}$. It suffices to show that the hypotheses parameterized by $x_{\frac{(i-1)T}{n+1}+1:\frac{iT}{n+1}}$ are also consistent with

$$\left(x_{\frac{(m-1)T}{n+1}+1}, y_{\frac{(m-1)T}{n+1}+1}\right), ..., \left(x_{\frac{mT}{n+1}}, y_{\frac{mT}{n+1}}\right).$$

To show this, recall that $j_m$ is the index computed in Lines 11-14 of Algorithm 9 on round $m$. Let

$$\left(\tilde{x}_{\frac{(m-1)T}{n+1}+1}, \tilde{y}_{\frac{(m-1)T}{n+1}+1}\right), ..., \left(\tilde{x}_{\frac{mT}{n+1}}, \tilde{y}_{\frac{mT}{n+1}}\right).$$

be the sample sorted in increasing order by examples. There are three cases to consider. Suppose $j_m = 1$, then $\tilde{y}_{\frac{(m-1)T}{n+1}+1} = 0$, and it must be the case that $\tilde{y}_{\frac{(m-1)T}{n+1}+p} = 0$ for all $p \in \{2, ..., \frac{T}{n+1}\}$. Since $x_{1:\frac{mT}{n+1}} = \mathrm{SG}(S^0, (j_1, ..., j_{m-1}))$, by definition of Algorithm 6, we have that the last $\frac{T}{n+1}$ entries of $\mathrm{SG}(S^0, (j_1, ..., j_m))$ all lie strictly to the left of $\tilde{x}_{\frac{(m-1)T}{n+1}+1}$. Moreover, by Observation 4, this is true of the last $\frac{T}{n+1}$ entries of $\mathrm{SG}(S^0, (j_1, ..., j_m, q_{m+1}, ..., q_{i-1}))$ for any $q_{m+1}, ..., q_{i-1} \in \{1, ...., \frac{T}{n+1}+1\}$. Therefore, we must have that $x_{\frac{(i-1)T}{n+1}+1:\frac{iT}{n+1}}$, which are the last $\frac{T}{n+1}$ entries of $\mathrm{SG}(S^0, (j_1, ..., j_{i-1}))$, lies strictly to the left of $\tilde{x}_{\frac{(m-1)T}{n+1}+1}$, implying that their associated hypotheses output 0 on all of $\left(x_{\frac{(m-1)T}{n+1}+1}, y_{\frac{(m-1)T}{n+1}+1}\right), ..., \left(x_{\frac{mT}{n+1}}, y_{\frac{mT}{n+1}}\right)$ as needed. By symmetry, when $j_m = \frac{T}{n+1} + 1$, we have that $x_{\frac{(i-1)T}{n+1}+1:\frac{iT}{n+1}}$ lies strictly to the right of $\tilde{x}_{\frac{mT}{n+1}}$, implying that their associated hypotheses output 1 on all of $\left(x_{\frac{(m-1)T}{n+1}+1}, y_{\frac{(m-1)T}{n+1}+1}\right), ..., \left(x_{\frac{mT}{n+1}}, y_{\frac{mT}{n+1}}\right)$ as needed. Now, consider the case where $j_m \in \{2, ..., \frac{T}{n+1}\}$. Then, by Algorithm 6 and Observation 4, for any $q_{m+1}, ..., q_i \in \{1, ...., \frac{T}{n+1}+1\}$, the last $\frac{T}{n+1}$ entries of $\mathrm{SG}(S^0, (j_1, ..., j_m, q_{m+1}, ..., q_{i-1}))$ lie strictly in between $\tilde{x}_{\frac{(m-1)T}{n+1}+j_m-1}$ and $\tilde{x}_{\frac{(m-1)T}{n+1}+j_m}$. Thus, the hypotheses parameterized by $x_{\frac{(i-1)T}{n+1}+1:\frac{iT}{n+1}}$ output 1 on examples $\tilde{x}_{\frac{(m-1)T}{n+1}+1:\frac{(m-1)T}{n+1}+j_m-1}$ and 0 on examples $\tilde{x}_{\frac{(m-1)T}{n+1}+j_m:\frac{mT}{n+1}}$. Finally, note that by definition of $j_m$, it must be the case that $\tilde{y}_{\frac{(m-1)T}{n+1}+1:\frac{(m-1)T}{n+1}+j_m-1} = (1, ..., 1)$ and $\tilde{y}_{\frac{(m-1)T}{n+1}+j_m:\frac{mT}{n+1}} = (0, ..., 0)$. Thus, once again the hypotheses parameterized by $x_{\frac{(i-1)T}{n+1}+1:\frac{iT}{n+1}}$ are consistent with the sample

$$\left(x_{\frac{(m-1)T}{n+1}+1}, y_{\frac{(m-1)T}{n+1}+1}\right), ..., \left(x_{\frac{mT}{n+1}}, y_{\frac{mT}{n+1}}\right).$$

This shows that these hypotheses are contained in $V_{m+1}$, completing the induction step. $\qquad\square$

**Step 5: Completing the proof of Theorem 3.8**

We are now ready to complete the proof of Theorem 3.8, which follows from composing E.1, E.2, E.3, E.4, and E.5. Namely, Lemma E.1 and the discussion in Section 15 show that there exists a Predictor $\mathcal{P}$ such that for any learner $\mathcal{A}$ playing according to Protocol 2, Online Classification with Predictions is equivalent to Online Classification with Peeks whenever the stream $(x_1, y_1), ..., (x_T, y_T)$ selected by the adversary satisfies the constraint that $x_{1:T} \in \mathcal{Z}_n$. Lemmas E.3 and E.4 show that for any learner $\mathcal{A}$, Nature playing according to Algorithm 9 guarantees that the resulting sequence $(x_1, y_1), ..., (x_T, y_T)$ satisfies the constraint that $x_{1:T} \in \mathcal{Z}_n$ and realizability by $\mathcal{H}$. Thus, for the Predictor $\mathcal{P}$ specified by Algorithm 7 and Nature playing according to Algorithm 9, Online Classification with Predictions is equivalent to Online Classification with Peeks. Finally, for Online Classification with Peeks, combining Lemmas E.2 and E.5 shows that for any learner $\mathcal{A}$, Nature, by playing according to Algorithm 9, guarantees that $\mathcal{A}$ makes at least $\frac{\log_2(\frac{T}{n+1})}{2}$ mistakes in expectation every $\frac{T}{n+1}$ rounds. Thus, Nature forces $\mathcal{A}$ to make at least $\frac{(n+1)}{2} \log_2(\frac{T}{n+1})$ mistakes in expectation by the end of the game, completing the proof.

# F  Adaptive Rates in the Agnostic Setting

In this section, we consider the harder agnostic setting and prove analogous results as in Section 3. Our main quantitative result is the agnostic analog of Theorem 3.1.

**Theorem F.1** (Agnostic upper bound). *For every $\mathcal{H} \subseteq \mathcal{Y}^{\mathcal{X}}$, Predictor $\mathcal{P}$, and no-regret offline learner $\mathcal{B}$, there exists an online learner $\mathcal{A}$ such that for every stream $(x_1, y_1), ..., (x_T, y_T)$, $\mathcal{A}$'s expected regret is at most*

$$
\bigg(\underbrace{\sqrt{\mathrm{L}(\mathcal{H})\, T \log_2(eT)}}_{(i)} \;\wedge\; \underbrace{\bigg(2(\mathrm{M}_{\mathcal{P}}(x_{1:T}) + 1)\, \overline{\mathrm{R}}_{\mathcal{B}}\Big(\frac{T}{\mathrm{M}_{\mathcal{P}}(x_{1:T}) + 1} + 1, \mathcal{H}\Big) + \sqrt{T \log_2 T}\bigg)}_{(ii)}\bigg) + \sqrt{T}.
$$

With respect to learnability, Corollary F.2 shows that offline learnability of $\mathcal{H}$ is sufficient for online learnability under predictable examples.

**Corollary F.2** (Offline learnability $\implies$ Agnostic Online learnability with Predictable Examples). *For every $\mathcal{H} \subseteq \mathcal{Y}^{\mathcal{X}}$ and $\mathcal{Z} \subseteq \mathcal{X}^{\star}$,*

$\mathcal{Z}$ *is predictable* and $\mathcal{H}$ *is offline learnable* $\implies$ $(\mathcal{H}, \mathcal{Z})$ *is agnostic online learnable.*

In addition, we can also establish a quantitative version of Corollary F.2 for VC classes.

**Corollary F.3.** *For every $\mathcal{H} \subseteq \{0, 1\}^{\mathcal{X}}$, Predictor $\mathcal{P}$, $\mathcal{Z} \subseteq \mathcal{X}^{\star}$, and no-regret offline learner $\mathcal{B}$, there exists an online learner $\mathcal{A}$ such that*

$$
\mathrm{R}_{\mathcal{A}}(T, \mathcal{H}, \mathcal{Z}) = O\bigg((\mathrm{M}_{\mathcal{P}}(T, \mathcal{Z}) + 1)\sqrt{\frac{\mathrm{VC}(\mathcal{H})\, T}{\mathrm{M}_{\mathcal{P}}(T, \mathcal{Z}) + 1} \log_2\Big(\frac{T}{\mathrm{M}_{\mathcal{P}}(T, \mathcal{Z}) + 1}\Big)} + \sqrt{T \log_2 T}\bigg).
$$

The proof of Corollary F.3 is in Section F.4. The remainder of this section is dedicated to proving Theorem 3.1 and Corollary F.3. The proof is similar to the realizable case. It involves constructing two online learners with expected regret bounds (i) and (ii) respectively, and then running the celebrated Randomized Exponential Weights Algorithm (REWA) using these learners as experts [Cesa-Bianchi and Lugosi, 2006]. The following guarantee of REWA along with upper bound (i) and (ii) gives the upper bound in Theorem F.1.

**Lemma F.4** (REWA guarantee [Cesa-Bianchi and Lugosi, 2006]). *The expected regret of* REWA *when run with $N$ experts and learning rate $\eta = \sqrt{\frac{8 \ln N}{T}}$ is at most $\min_{i \in [N]} M_i + \sqrt{T \log_2 N}$, where $M_i$ is the number of mistakes made by expert $i \in [N]$.*

The online learner obtaining the regret bound $\sqrt{\mathrm{L}(\mathcal{H})\, T \log_2(eT)}$ is the generic agnostic online learner from Hanneke et al. [2023], thus we omit the details here. Our second learner is described in Section F.2 and uses Algorithm 3 as a subroutine. The following lemma, bounding the expected regret of Algorithm 3 in the agnostic setting, will be crucial.

**Lemma F.5.** *For every $\mathcal{H} \subseteq \mathcal{Y}^{\mathcal{X}}$, Predictor $\mathcal{P}$, no-regret offline learner $\mathcal{B}$, and stream $(x_1, y_1), ..., (x_T, y_T)$, the expected regret of Algorithm 3 is at most $(\mathrm{M}_{\mathcal{P}}(x_{1:T}) + 1)\, \mathrm{R}_{\mathcal{B}}(T, \mathcal{H})$.*

### F.1 Proof of Lemma F.5

The proof closely follows that of Lemma 3.5.

*Proof.* Let $\mathcal{A}$ denote Algorithm 3 and $(x_1, y_1), ..., (x_T, y_T)$ denote the stream to be observed by $\mathcal{A}$. Let $c$ be the random variable denoting the number of mistakes made by Predictor $\mathcal{P}$ on the stream and $t_1, ..., t_c$ be the random variables denoting the time points where $\mathcal{P}$ makes these errors (e.g . $\hat{x}_{t_i}^{t_i - 1} \neq x_{t_i}$). Note that $t_i \geq 2$ for all $i \in [c]$. We will show pointwise for every value of $c$ and $t_1, ..., t_c$ that $\mathcal{A}$ makes at most $(c + 1)\, \mathrm{R}_{\mathcal{B}}(T, \mathcal{H})$ mistakes in expectation over the randomness of $\mathcal{B}$. Taking an outer expectation with respect to the randomness of $\mathcal{P}$ and using the fact that $\mathbb{E}[c] = \mathrm{M}_{\mathcal{P}}(x_{1:T})$, completes the proof.

First, consider the case where $c = 0$ (i.e. $\mathcal{P}$ makes no mistakes). Then, since $\mathcal{P}$ is lazy, we have that $\hat{x}_{1:T}^{t} = x_{1:T}$ for every $t \in [T]$. Thus line 5 fires exactly once on round $t = 1$, $\mathcal{A}$ initializes an offline learner $\mathcal{B}^1$ with $x_{1:T}$, and $\mathcal{A}$ uses $\mathcal{B}^1$ to make its prediction on all rounds. Thus, $\mathcal{A}$ makes at most $\mathrm{R}_{\mathcal{B}}(T, \mathcal{H})$ mistakes in expectation.

Now, let $c > 0$ and $t_1, ..., t_c$ be the time points where $\mathcal{P}$ errs. Partition the sequence $1, ..., T$ into the disjoint intervals $(1, ..., t_1 - 1), (t_1, ..., t_2 - 1), ..., (t_c, ..., T)$. Define $t_0 := 1$ and $t_{c+1} := T$. Fix an

$i \in \{0, ..., c\}$. Then, for every $j \in \{t_i, ..., t_{i+1} - 1\}$, we have that $\hat{x}^j_{1:t_{i+1}-1} = x_{t_{i+1}-1}$. This comes from the fact that $\mathcal{P}$ does not error on timepoints $t_i + 1, ..., t_{i+1} - 1$ and is both consistent and lazy (see Assumptions 1 and 2). Thus, line 5 fires on round $t_i$, $\mathcal{A}$ initializes an offline learner $\mathcal{B}^i$ with the sequence $\hat{x}^{t_i}_{t_i:T} = x_{t_i:t_{i+1}-1} \circ \hat{x}^{t_i}_{t_{i+1}:T}$, and $\mathcal{A}$ uses $\mathcal{B}^i$ it to make predictions for all remaining timepoints $t_i, ..., t_{i+1} - 1$. Note that line 5 does not fire on timepoints $t_i + 1, ..., t_{i+1} - 1$.

Let $h^i \in \arg\min_{h \in \mathcal{H}} \sum_{t=t_i}^{t_{i+1}-1} \mathbb{1}\{h(x_t) \neq y_t\}$ be an optimal hypothesis for the partition $(t_i, ..., t_{i+1} - 1)$. Let $y^i_t = y_t$ for $t_i \leq t \leq t_{i+1} - 1$ and $y^i_t = h^i(\hat{x}^{t_i}_t)$ for all $t \geq t_{i+1}$. Then, note that

$$\inf_{h \in \mathcal{H}} \sum_{t=t_i}^{T} \mathbb{1}\{h(\hat{x}^{t_i}_t) \neq y^i_t\} = \inf_{h \in \mathcal{H}} \sum_{t=t_i}^{t_{i+1}-1} \mathbb{1}\{h(x_t) \neq y_t\}.$$

Now, consider the hypothetical labeled stream

$$(\hat{x}^{t_i}_{t_i}, y^i_{t_i}), ..., (\hat{x}^{t_i}_T, y^i_T) = (x_{t_i}, y_{t_i}), ..., (x_{t_{i+1}-1}, y_{t_{i+1}-1}), (\hat{x}^{t_i}_{t_{i+1}}, y^i_{t_{i+1}}), ..., (\hat{x}^{t_i}_T, y^i_T).$$

By definition, $\mathcal{B}^i$, after initialized with $\hat{x}^{t_i}_{t_i:T}$, makes at most

$$\inf_{h \in \mathcal{H}} \sum_{t=t_i}^{T} \mathbb{1}\{h(\hat{x}^{t_i}_t) \neq y^i_t\} + \mathrm{R}_\mathcal{B}(T - t_i, \mathcal{H}) = \inf_{h \in \mathcal{H}} \sum_{t=t_i}^{t_{i+1}-1} \mathbb{1}\{h(x_t) \neq y_t\} + \mathrm{R}_\mathcal{B}(T - t_i, \mathcal{H})$$

mistakes in expectation when simulated on the stream $(\hat{x}^{t_i}_{t_i}, y^i_{t_i}), ..., (\hat{x}^{t_i}_T, y^i_T)$. Thus, $\mathcal{B}^i$ makes at most $\inf_{h \in \mathcal{H}} \sum_{t=t_i}^{t_{i+1}-1} \mathbb{1}\{h(x_t) \neq y_t\} + \mathrm{R}_\mathcal{B}(T - t_i + 1, \mathcal{H})$ mistakes in expectation on the *prefix* $(\hat{x}^{t_i}_{t_i}, y^i_{t_i}), ..., (\hat{x}^{t_i}_{t_{i+1}-1}, y^i_{t_{i+1}-1}) = (x_{t_i}, y_{t_i}), ..., (x_{t_{i+1}-1}, y_{t_{i+1}-1})$. Since on timepoint $t_i$, $\mathcal{A}$ instantiates $\mathcal{B}^i$ with the sequence $\hat{x}^{t_i}_{t_i:T}$ and proceeds to simulate $\mathcal{B}^i$ on the sequences of labeled examples $(x_{t_i}, y_{t_i}), ..., (x_{t_{i+1}-1}, y_{t_{i+1}-1})$, $\mathcal{A}$ makes at most $\inf_{h \in \mathcal{H}} \sum_{t=t_i}^{t_{i+1}-1} \mathbb{1}\{h(x_t) \neq y_t\} + \mathrm{R}_\mathcal{B}(T - t_i + 1, \mathcal{H})$ mistakes in expectation on the sequence $(x_{t_i}, y_{t_i}), ..., (x_{t_{i+1}-1}, y_{t_{i+1}-1})$. Since the interval $i$ was chosen arbitrarily, this is true for every $i \in \{0, ..., c\}$ and

$$\mathbb{E}\left[\sum_{t=1}^{T} \mathbb{1}\{\mathcal{A}(x_t) \neq y_t\}\right] = \mathbb{E}\left[\sum_{i=0}^{c} \sum_{t=t_i}^{t_{i+1}-1} \mathbb{1}\{\mathcal{A}(x_t) \neq y_t\}\right]$$

$$\leq \sum_{i=0}^{c} \left(\inf_{h \in \mathcal{H}} \sum_{t=t_i}^{t_{i+1}-1} \mathbb{1}\{h(x_t) \neq y_t\} + \mathrm{R}_\mathcal{B}(T - t_i + 1, \mathcal{H})\right)$$

$$\leq \inf_{h \in \mathcal{H}} \sum_{t=1}^{T} \mathbb{1}\{h(x_t) \neq y_t\} + (c+1)\mathrm{R}_\mathcal{B}(T, \mathcal{H}),$$

as needed. □

## F.2 Proof of upper bound (ii) in Theorem F.1

The proof of upper bound (ii) in Theorem F.1 closely follows the proof of upper bound (iii) in Theorem 3.1 from the realizable setting. The main idea is to run REWA using the same experts defined in Algorithm 4 and bounding the expected regret in terms of the expected regret of $\mathcal{K}$ from Lemma F.5.

We show that Algorithm 5 using REWA in line 3 and the experts in Algorithm 4 with their guarantee in Lemma F.5 achieves upper bound (ii) in Theorem F.1. Let $(x_1, y_1), ..., (x_T, y_T)$ be the stream to be observed by the learner. Let $\mathcal{A}$ denote the online learner in Algorithm 5 using REWA in line 3 of

Algorithm 5. By the guarantees of the REWA, we have that

$$\mathbb{E}\left[\sum_{t=1}^{T} \mathbb{1}\{\mathcal{A}(x_t) \neq y_t\}\right] \leq \mathbb{E}\left[\inf_{b \in \{0,...,T-1\}} \sum_{t=1}^{T} \mathbb{1}\{E_b(x_t) \neq y_t\}\right] + \sqrt{T \log_2 T}$$

$$\leq \mathbb{E}\left[\sum_{t=1}^{T} \mathbb{1}\{E_{\lceil M_{\mathcal{P}}(x_{1:T})\rceil}(x_t) \neq y_t\}\right] + \sqrt{T \log_2 T}$$

$$\leq \mathbb{E}\left[\sum_{i=0}^{\lceil M_{\mathcal{P}}(x_{1:T})\rceil} \sum_{t=\tilde{t}_i+1}^{\tilde{t}_{i+1}} \mathbb{1}\{\mathcal{K}_i(x_t) \neq y_t\}\right] + \sqrt{T \log_2 T}$$

$$\leq \mathbb{E}\left[\sum_{i=0}^{\lceil M_{\mathcal{P}}(x_{1:T})\rceil} (m_i+1)\overline{R}_{\mathcal{B}}(\tilde{t}_{i+1} - \tilde{t}_i, \mathcal{H}) + \inf_{h \in \mathcal{H}} \sum_{t=\tilde{t}_i+1}^{\tilde{t}_{i+1}} \mathbb{1}\{h(x_t) \neq y_t\}\right] + \sqrt{T \log_2 T}$$

$$\leq \mathbb{E}\left[\sum_{i=0}^{\lceil M_{\mathcal{P}}(x_{1:T})\rceil} (m_i+1)\overline{R}_{\mathcal{B}}(\tilde{t}_{i+1} - \tilde{t}_i, \mathcal{H})\right] + \inf_{h \in \mathcal{H}} \sum_{t=1}^{T} \mathbb{1}\{h(x_t) \neq y_t\} + \sqrt{T \log_2 T}$$

$$\leq \inf_{h \in \mathcal{H}} \sum_{t=1}^{T} \mathbb{1}\{h(x_t) \neq y_t\} + 2(M_{\mathcal{P}}(x_{1:T})+1)\overline{R}_{\mathcal{B}}\left(\frac{T}{M_{\mathcal{P}}(x_{1:T})+1} + 1, \mathcal{H}\right) + \sqrt{T \log_2 T},$$

where the fourth inequality uses the guarantee of $\mathcal{K}$ from Lemma F.5 and the last inequality follows using an identical argument as in the proof of Lemma 3.6 since $\overline{R}_{\mathcal{B}}(T, \mathcal{H})$ is a concave, sublinear function of $T$.

### F.3 Proof of Theorem F.1

Let $\mathcal{A}$ denote the REWA using the generic agnostic online learner from Hanneke et al. [2023] and the algorithm described in Section F.2 as experts. Then, for any stream $(x_1, y_1), ..., (x_T, y_T)$, Lemma F.4 gives that

$$\mathbb{E}\left[\sum_{t=1}^{T} \mathbb{1}\{\mathcal{A}(x_t) \neq y_t\}\right] \leq \mathbb{E}\left[\min_{i \in [2]} M_i\right] + \sqrt{T} \leq \min_{i \in [2]} \mathbb{E}[M_i] + \sqrt{T},$$

where we take $M_1$ and $M_2$ to be the number of mistakes made by the generic agnostic online learner from Hanneke et al. [2023] and the algorithm described in Section F.2 respectively. Note that $M_1$ and $M_2$ are random variables. Finally, using [Hanneke et al., 2023, Theorem 4] as well as upper bound (ii) completes the proof of Theorem F.1.

### F.4 Proof of Corollaries F.2 and F.3

The proof of the generic upper bound on $M_{\mathcal{A}}(T, \mathcal{H}, \mathcal{Z})$ follows by using the same learner $\mathcal{A}$ as in the proof of upper bound (ii) in Theorem F.1. However, this time we bound

$$\mathbb{E}\left[\inf_{b \in \{0,...,T-1\}} \sum_{t=1}^{T} \mathbb{1}\{E_b(x_t) \neq y_t\}\right] \leq \mathbb{E}\left[\sum_{t=1}^{T} \mathbb{1}\{E_{\lceil M_{\mathcal{P}}(T, \mathcal{Z})\rceil}(x_t) \neq y_t\}\right]$$

and use an identical analysis as in the proof of upper bound (ii) and Lemma 3.6 to get

$$R_{\mathcal{A}}(T, \mathcal{H}, \mathcal{Z}) = O\left(\sqrt{L(\mathcal{H}) T \log_2 T} \wedge \left((M_{\mathcal{P}}(T, \mathcal{Z})+1)\overline{R}_{\mathcal{B}}\left(\frac{T}{M_{\mathcal{P}}(T, \mathcal{Z})+1}, \mathcal{H}\right) + \sqrt{T \log_2 T}\right)\right).$$

Corollary F.2 follows from the fact that $R_{\mathcal{A}}(T, \mathcal{H}, \mathcal{Z}) = o(T)$ if $M_{\mathcal{P}}(T, \mathcal{Z}) = o(T)$ and $R_{\mathcal{B}}(T, \mathcal{H}) = o(T)$. To get the upper bound in Corollary F.3, it suffices to plug in the upper bound $\overline{R}_{\mathcal{B}}(T, \mathcal{H}) = O\left(\sqrt{VC(\mathcal{H}) T \log_2 T}\right)$, given by Theorem 6.1 from Hanneke et al. [2024], into the above upper bound on $R_{\mathcal{A}}(T, \mathcal{H}, \mathcal{Z})$.

