# OpenReview forum: "Online Classification with Predictions"
_NeurIPS.cc/2024/Conference — NeurIPS 2024 poster_

### Official Review · Reviewer_f4Qh · 2024-06-25

**Soundness:** 3
**Presentation:** 2
**Contribution:** 3
**Rating:** 6
**Confidence:** 3

**Summary:**

The paper studies an online classification problem that interpolates between transductive online classification and (standard) online classification. The interpolation is controlled by a "predictor" entity that tries to predict the examples to be labelled: If the predictor is good, the setting is more like transductive online classification, and otherwise it is more like (standard) online classification. For this problem, the authors design an online learner that uses a predictor and can do better than a standard online learning algorithm if the predictor is good. As a corollary, they show that transductive online learning implies online learning when the input stream is taken from a "predictable" collection of streams. For the first result, they also provide a lower bound that shows that it is nearly tight in some sense.

**Strengths:**

1. The paper presents a new and fairly natural problem in online learning, and provides nearly tight guarantees for it.
2. The idea of using a few learners to cover all cases and then use WM on them is useful and interesting (is it novel?).

**Weaknesses:**

1. Most of the proofs use combinations of known techniques.
2. There is some room for improvements in the writing and presentation. For example, in page 2, I didn't understand the difference between contributions (1) and (2).

**Questions:**

Questions:
1. To provide motivation, it is written that the standard online learning setting is often too pessimistic. However, there are other ways to make it more optimistic. For example, what about a stochastic adversary? It seems like a more natural way to make the setting more optimistic, but it is not discussed in the paper.
2. Why is line 172 "without loss of generality"?
3. I am not quite sure that the story of a "predictor" in the background is the most natural way to present the problem. Isn't the setting you define is just transductive online learning with possible noise in the given examples (which is interesting as well)?
4. A notion of "predictable stream collections" is defined, but the paper does not study it or even just discuss open questions about it. Do we know how a predictable collection of streams looks like?


Suggestions:
1. I didn't understand the sentence in lines 155-156. I suggest rephrasing.
2. The weighted majority algorithm is originally due to the seminal work [1]. I would mention it.

Typos/english:
1. Line 56 is unclear.
2. Line 145: "to ability to define"

References:

[1] Nick Littlestone and Manfred K Warmuth. The weighted majority algorithm.Information and computation, 108(2):212–261, 1994.

**Limitations:**

Yes

---

> ### Author Rebuttal · Authors · 2024-08-03
>
> We thank the reviewer for noting that we present a new and fairly natural problem in online learning. We will incorporate all suggestions and fix typos in the final camera-ready version. We address each weakness and question below.
>
> **Strengths**
>
> 2. This technique of constructing a few online learners and running MW on top is not completely new. For example, this technique is used to design agnostic online learning algorithms for multiclass and binary online classification [1, 2]. However, in the context of Online Algorithms with Predictions, we are unaware of any other work that uses this technique.
>
> **Weaknesses**
> 1. Although our techniques may not be groundbreaking, we disagree with the reviewer that our proofs use combinations of existing ones. Since this setting has not been studied before, we argue that new techniques needed to be developed. For example, it is not obvious apriori how to construct the class of Experts used by Algorithm 5 or that this entire approach as a whole leads to a near optimal upper bound.
> 2. We thank the reviewer for pointing out the ambiguity between contributions (1) and (2).  Contribution (1) is mainly aimed at quantifying the minimax rates while Contribution (2) is concerned with characterizing learnability. In particular, Contribution (1) aims to answer: For any hypothesis class $H$ and a predictor $P$, what is the minimax expected number of mistakes/regret when given access to $P$? Here, the goal is to derive mistake and regret bounds in terms of the performance of $P$.  Contribution (2) aims to answer: does having access to a Predictor make online learning easier and if so, when? That is, are there classes $H$ that are online learnable when given access to a Predictor but not online learnable otherwise?  If so, which classes $H$ become online learnable when given access to a Predictor? We will make this distinction more clear in the final version.
>
> **Questions**
> 1. There are indeed other ways of making the standard online learner setting more optimistic. The "stochastic adversary" mentioned by the reviewer is actually one of them and is known formally as "smoothed online learning" in literature. We do mention smoothed online learning in Line 64-65 and Appendix A as an alternative way of going beyond a worst-case analysis. We will add a larger discussion about the smoothed model in the main text of the final version.
> 2. Since realizable and agnostic learnability are equivalent, it doesn't matter whether we use either (1) the existence of a learner with sublinear mistake bound or (2) the existence of a learner with sublinear regret to characterize offline learnability in both the realizable or agnostic settings. Lines 172-173 is just saying that we will always pick  (2) the existence of a learner with sublinear regret, to characterize offline learnability in both the realizable and agnostic settings. We will remove this phrase in the final version to avoid confusion.
> 3. Our setting with a Predictor is actually more general than "transductive online learning with possible noise in the given examples." This is because we allow the Predictor to change its predictions about future instances throughout the game. On the other hand, in the noisy transductive online setting, one presumably fixes the noisy instances revealed to the learner before the game begins. We give a brief discussion of this on lines 136-141. In addition, we find that access to a Predictor (and therefore dynamic predictions about future instances) is more natural in practice. For example, one often has the ability to update predictions about future instances given the present instance (e.g. temperature forecasting).
> 4. One natural predictable collection of streams are those induced by dynamical systems. That is, let $X$ be the state space for a collection of transition functions $G: X \mapsto X$. Then, given an initial state $x_0$, one can consider the stream class $Z$ to be the set of all trajectories induced by transition functions in $G$. We will add a discussion of what predictable stream classes might look like in the camera ready version.
>
>
> [1] Ben-David, Shai, Dávid Pál, and Shai Shalev-Shwartz. "Agnostic Online Learning." COLT. Vol. 3. 2009.
>
> [2] Hanneke, Steve, et al. "Multiclass online learning and uniform convergence." The Thirty Sixth Annual Conference on Learning Theory. PMLR, 2023.

---

> > ### Comment · Reviewer_f4Qh · 2024-08-10
> >
> > Thank you very much for addressing my comments and questions.

---

### Official Review · Reviewer_HoBJ · 2024-07-09

**Soundness:** 3
**Presentation:** 3
**Contribution:** 3
**Rating:** 6
**Confidence:** 2

**Summary:**

This paper studies the complexity of the online classification problem when provided with a predictor $\mathcal{P}$ that can forecast future features of data samples. Using black-box access to this predictor, the authors provide a beyond-worst-case analysis of online classification algorithms that can adapt to the 'easiness' of the online stream. Specifically, in Theorem 3.1, they demonstrate that there exists an online learner whose mistakes can be upper bounded by the Littlestone dimension, while additionally being bounded by the performance of predictor $\mathcal{P}$ and that of the employed offline learner. As a corollary, their result reveals the relationship between offline learnability and realizable online learnability given such a predictor. A lower bound is provided to demonstrate the tightness of the upper bound.

**Strengths:**

1. The presentation of this paper is clear, and I can quickly grasp the main idea of how to construct an online learner that satisfies the upper bound.

2. This paper explores beyond-worst-case guarantees by leveraging a predictor and an offline learner. With the constructed online protocol, the authors reveal the linkage between offline learnability and online learnability to some extent.

3. The theoretical results appear solid with both the upper bounds and the lower bound, although I have not checked the appendices carefully.

**Weaknesses:**

My main concern is about the accuracy of the predictor $\mathcal{P}$. Unlike in the offline setting, where the features $x_{1:T}$ are naturally provided, in the online setting—especially the adversarial one—the predictor $\mathcal{P}$ might not be accurate, and the factor $M_{\mathcal{P}}$ could be extremely large. Also, Algorithm 3 will initialize a new copy every time the predictor makes a mistake. It seems that Algorithm 3 might initialize the copy frequently, and it might not learn the potential pattern even if the mistakes made are minor.

**Questions:**

Could we consider 'soft' criteria when initializing the copy? For example, could the online learner initialize a new copy only when the cumulative errors, such as $\sum_t \|x_t - \hat{x}_t\|$, exceed some threshold?

---

> ### Author Rebuttal · Authors · 2024-08-03
>
> We thank the reviewer for finding the theoretical results to be solid and the presentation to be clear. We address the concerns below.
>
> - Indeed, the stream of instances $x_1, ..., x_T$ could be very unpredictable and $M_P$ can be very large. However, in this case the minimum in Theorem 3.1 is obtained by $L(H)$, the Littlestone dimension of $H$. Thus, our algorithm's guarantee is never worse than the worst-case mistake bound (up to constant factors). However, if $M_P$ is very small, then our algorithm's guarantee (see (ii) and (iii) in Theorem 3.1) can be much smaller than $L(H)$. In this way, our algorithm adapts to the quality of predictions made by $P$.
>
> - Indeed, if the Predictor makes a lot of mistakes, Algorithm 3 will restart the Offline learner frequently causing its expected number of mistakes to explode. This is precisely why we give Algorithm 5, which explicitly controls the number of restarts. Note that the expected number of mistakes made by Algorithm 5 (see Lemma 3.7) can be significantly smaller than that by Algorithm 3 (see Lemma 3.5), and this is captured by term (iii) in Theorem 3.1.
>
> - The "soft" criteria when initializing a new copy is an interesting idea. However, it is not immediately obvious to us that such an idea would work since the Offline learner is only useful when the instances it gets initialized with actually matches the true instances in the stream. In any case, note that the guarantee of Algorithm 5 (see Lemma 3.7) is already near optimal (up to log factors in $T$)  given the lower bound in Theorem 3.8.

---

> > ### Comment · Reviewer_HoBJ · 2024-08-09
> >
> > Thanks for the authors' replies, which address my concerns. So I decide to raise my score to 6.

---

### Official Review · Reviewer_yULk · 2024-07-13

**Soundness:** 4
**Presentation:** 3
**Contribution:** 3
**Rating:** 7
**Confidence:** 3

**Summary:**

The paper belongs to the field of statistical learning. The learner is seeing pairs $(x,y)$ and needs to predict labels $y$s to compete (in terms of the expected number of errors) with any hypothesis from the given class.

The main result of the paper, Theorem 3.1, establishes a connection between the offline and the online setup. In the offline setup, the learner knows the sequence $x_1, x_2, \ldots, x_T$ where it needs to predict labels in advance. In the online setup it gets $x$s one by one but gets help from a predictor $\cal P$ that tries to predict what remains of the sequence. The paper constructs an online algorithm utilising an offline algorithm and a predictor and gets an upper bound on its performance in terms of the quality of the offline bound and the quality of the predictor.

Corollary 3.2 shows that if a class of sequences is predictable ( = can be predicted with sublinear cumulative error) and a class of hypotheses is learnable under offline settings (the regret is sublilear), then their combination is learnable under online settings.

Theorem 3.8 provides a lower bound.

**Strengths:**

I believe the paper presents an interesting result answering an important question.

**Weaknesses:**

I am worried the very technical presentation may limit the appeal of the paper.

Typos etc:

Page 4, Section 2.3, first line: to ability -> the ability

Page 4, Section 2.3. I believe the mentioning of the adversary appears here for the first time, and it is confusing. I understand we are talking of the adversary because we are interested in the supremum of the loss over sequences, but I may be wrong...

Page 6. Lemma 2.2 says there is _a_ concave function upper-bounding $g$. Then $\bar f$ is defined as _the_ concave function upper-bounding $f$. Which one is used? The minimal perhaps?

**Questions:**

None

**Limitations:**

Yes

---

> ### Author Rebuttal · Authors · 2024-08-03
>
> We thank the reviewer for finding that our work presents an "interesting result answering an important question." We address the concerns below.
>
> - We thank the reviewer for pointing out our use of "adversary" in Page 4 Section 2.3. We will change this to "Nature" to make it consistent  with the setup in Section 2.1.
> - Yes, we take \bar{f} to the smallest concave sublinear function that upper bounds g. We will make this explicit in the camera-ready version.

---

### Official Review · Reviewer_EJB4 · 2024-07-15

**Soundness:** 4
**Presentation:** 3
**Contribution:** 3
**Rating:** 6
**Confidence:** 3

**Summary:**

This paper studies learning-augmented online classification, where the classifier has access to a predictor that forecasts future examples. The paper proposes an algorithm that uses these predictions, and the algorithm's performance depends on the prediction error. The proposed algorithm is robust (never worse than the worst-case mistake bound), consistent (as good as the best offline learner if the predictions are perfect), and degrades smoothly as prediction quality decreases. Their findings show that accurate predictions can make online learning as straightforward as transductive online learning. The contribution of this paper is theoretical.

**Strengths:**

- This is the first paper exploring the task of online classification with predictions. The main result is intuitive and elegant, showing that "online learnability with an effective predictor can be reduced to transductive online learnability."
- The theoretical part of this paper is sound and well-organized. The paper effectively analyzes the robustness, consistency, and smoothness of the proposed algorithm, which are key aspects of learning-augmented algorithms.

**Weaknesses:**

- The paper does not discuss how the predictions can be generated, particularly from machine learning models. Admittedly, most designs in learning-augmented algorithms focus on the algorithmic side. Yet, linking the proposed algorithm to settings where machine learning models can be involved would strengthen the paper's relevance to the NeurIPS audience.

- The paper does not include any experiment or evaluation on how the proposed algorithms would perform in real-world scenario. Similar to the previous point, I think the community studying learning-augmented algorithms value both theoretical soundness and practicality (which is the reason to include  learned predictors in the first place). I think some experiments (even on synthetic data) would make this paper more complete.
- The paper lacks experiments or evaluations on the proposed algorithms' performance in real-world scenarios.  Similar to the previous point, I think the community studying learning-augmented algorithms value both theoretical soundness and practicality (which is the reason to include  learned predictors in the first place). Some experiments even on synthetic data showcasing the improved performance of the proposed algorithm would make this paper more comprehensive.

**Questions:**

- I am not an expert in online classification, but are there any public benchmarks or test sets to evaluate the performance of online classification algorithms? If so, I strongly encourage the author to perform experiments to demonstrate the proposed algorithm's performance.
- In the third sentence of line 107, should it be $\hat y_t \in \mathcal{Y}$? Otherwise, I am confused about why the prediction is binary.
- In lines 377 - 380, do these two citations refer to the same paper?
- The notation in the paper is somewhat confusing. Specifically, regarding the predictor, I did not find the formal definition of $\mathcal{P}\left(x_{1: t}\right)$ and $\hat{x}_{1: T}$.

**Limitations:**

The majority of limitations are addressed in the 'weaknesses' and 'questions' sections. From a broader societal perspective, given the theoretical nature of this work, there is no immediate negative impact, and I cannot foresee any.

---

> ### Author Rebuttal · Authors · 2024-08-03
>
> We thank the reviewer for noting that our main result is intuitive and elegant. We address each of the weaknesses and questions below.
>
> **Weaknesses**
> - In this paper, we consider abstract instance spaces $X$. Accordingly, we did not discuss how the predictions can be generated since there is unlikely to be a one-size-fits-all Predictor. That said, for particular choices of the instance space $X$, existing literature in ML can certainly provide methods for generating predictions. For example, if $X$ is the state space for a discrete-time dynamical system, then existing algorithms for next-state predictions for discrete-time dynamical systems can be used to forecast future instances/states.  We will make sure to include a concrete example of how predictions may be generated from ML models in the camera ready version.
> - This paper is mainly focused on the theoretical/conceptual benefits of instance predictions. In particular, an important implication/insight of our results is that the difficulty in adversarial online classification is not due to the uncertainty of the labels, but rather, the uncertainty with respect to future instances. That said, we do agree with the reviewer that experiments on synthetic data showcasing the improved performance of the proposed algorithms is an interesting and important direction. We do note that our algorithms use black-box access to Offline learners and Predictors. Thus, the efficiency of our learning algorithms depend on the efficiency of existing Offline learner and Predictors. To the best of our knowledge, we do not know of any implementation of an Offline learner, let alone an efficient one. In particular, the Offline learner from Hanneke et al. [2023] is provable inefficiency since it requires computing the VC dimension.  Finally, we would like to point out that there are several influential papers in learning-augmented algorithms whose contributions were primarily theoretical [1, 2, 3, 4, 5].
>
> **Questions**
> - Yes, there are public benchmarks where one can evaluate the performance of online classification algorithms (e.g UCI ML Repository). However, these public benchmarks are also used to evaluate batch learning algorithms. Accordingly,  it is unclear how to design a Predictor for these datasets since these datasets are not inherently online - it is not known how the instances were generated or whether there is any particular order to the rows in the datasets.
> - Yes, in the third sentence of line 107, it should be $\hat{y}_t \in Y$.
> - Yes, these two citations cite the same paper. We will fix this in the final version.
> - We will make sure to define the Predictor and $\hat{x}^t_{1:T}$ explicitly in the camera-ready version. We use $P(x_{1:t})$ to denote the prediction made by Predictor $P$ after observing $x_1, ..., x_t$. We also use $\hat{x}^t_{1:T}$ to denote these predictions when it is more convenient. We will make sure to define these quantities explicitly.
>
>
> [1] Rohatgi, D. (2020). Near-optimal bounds for online caching with machine learned advice. In Proceedings of the Fourteenth Annual ACM-SIAM Symposium on Discrete Algorithms (pp. 1834-1845). Society for Industrial and Applied Mathematics.
>
> [2] Dütting, Paul, et al. "Secretaries with advice." Proceedings of the 22nd ACM Conference on Economics and Computation. 2021.
>
> [3] Lattanzi, Silvio, et al. "Online scheduling via learned weights." Proceedings of the Fourteenth Annual ACM-SIAM Symposium on Discrete Algorithms. Society for Industrial and Applied Mathematics, 2020.
>
> [4] Antoniadis, Antonios, et al. "Secretary and online matching problems with machine learned advice." Advances in Neural Information Processing Systems 33 (2020): 7933-7944.
>
> [5] Angelopoulos, Spyros, et al. "Online computation with untrusted advice." arXiv preprint arXiv:1905.05655 (2019).

---

> > ### Comment · Reviewer_EJB4 · 2024-08-14
> >
> > Thanks for the authors' replies and clarification. I will maintain my positive rating.

---

### Decision · Program_Chairs · 2024-09-25

**Decision:**

Accept (poster)

**Comment:**

The paper studies online classification when the learner has access to predictions about instances. For the Binary classification setting an online algorithm is derived which in the worst case attains the minimax regret bound against the worst case sequence of instances but at the same time, if the instances generated are realizable by a predictor in the function class, or in other words if the learner is guaranteed to observe data where future examples are easily predictable then the mistake bound obtained matches the online transductive setting (where Littlestone dimension is replaced by VC dimension).

The reviewers are in agreement that this paper is a strong theoretical paper. I agree as well and recommend an accept.